

# STITCHES: creating new scenarios of climate model output by stitching together pieces of existing simulations

Claudia Tebaldi[1,2], Abigail Snyder[2], and Kalyn Dorheim[2]

[1]Lawrence Berkeley National Laboratory, Berkeley, CA
[2]Joint Global Change Research Institute, Pacific Northwest National Laboratory and University of Maryland, College Park, MD

**Correspondence:** Claudia Tebaldi (ctebaldi@lbl.gov)

**Abstract.** Climate model output emulation has long been attempted to support impact research, mainly to fill-in gaps in the scenario space. Given the computational cost of running coupled Earth System Models (ESMs) an effective emulator would be used to create climatic impact-driver information under scenarios that could not be run by ESMs. Lately, the necessity of accounting for internal variability has also made the availability of initial condition ensembles important, increasing further

the computational demand. However, at least so far, emulators have always been limited to simplified ESM output, either seasonal, annual or decadal averages, and/or basic quantities, like temperature and precipitation, often emulated independently of one another. With this work, we propose a more comprehensive solution to climate model output emulation. Our emulator, STITCHES, uses existing archives of Earth System Models' (ESMs) scenario experiments to construct new scenarios, or enrich existing initial condition ensembles, which is what other emulators do. Importantly, its output has the same characteristics

of the ESM output it set out to emulate: multivariate, spatially resolved and high frequency as the original ESM output is. STITCHES extends the idea of time-sampling - by which climate outcomes are stratified by the global warming level at which they occur, irrespective of the scenario and time associated to them - to the construction of a continuous Global Surface Air Temperature (GSAT) trajectory over the whole 21[st] century that replicates a target trajectory to be emulated. STITCHES does so by stitching together decade-long windows within a model simulation when GSAT has similar characteristics to the target

GSAT trajectory, but in doing so STITCHES creates a series of pointers to a sequence of decades within existing scenarios in the ESM archived output, and the emulator can thus recover any type of output, at any frequency and spatial scale available from the original ESM's experiment that produced each decade. We show that the stitching does not introduce artifacts, in the great majority of cases, even when the criteria for the identification of the decades to be stitched together are not strictly tailored to the specific ESM emulated. We show this is the case for the variable that we expect to be smoother and less noisy

than many variables commonly used for impact analysis, annual GSAT. Our results also suggest that most other surface atmospheric variables commonly used for impact analysis would be similarly unaffected by the stitching procedure. We successfully test the method's performance over many CMIP6/ScenarioMIP-participating ESMs and experiments. Only a few exceptions surface, but these less-than-optimal outcomes are always associated with a scarcity of the archived simulations from which to gather the decade-long windows that form the emulated GSAT trajectory. In the great majority of cases, STITCHES perfor-

mance remains satisfactory according to metrics that reward consistency in trends, interannual and inter-ensemble variance,



and autocorrelation structure of the time series stitched together. The method therefore can be used to create new scenarios with different GSAT pathways than existing simulations, and to increase the size of existing initial condition ensembles. There are aspects of our emulator that will immediately disqualify it for specific applications, like when climate information is needed whose characteristics result from accumulated quantities over windows of times longer than those used as building blocks by STITCHES. But for many applications, we argue that a stitched product can satisfy the needs of impact researchers. Thus, we think it could open up the possibility of designing the next scenario experiments within CMIP7 according to new principles, relieved of the need to produce a number of similar trajectories that vary only in radiative forcing strength.

# 1   Introduction

In this paper, we introduce a novel and comprehensive solution to climate model emulation. Our principal motivation is to support the need by the impact research community for climate information under arbitrary future scenarios of anthropogenic forcings, but we believe that our proposal may potentially benefit the scenario development, integrated assessment and climate modeling communities.

The overarching problem that our method seeks to resolve stems from the computational and human labor costs of running climate model experiments according to plausible future scenarios (as opposed to idealized forcings, e.g., 1% $CO_2$ increase pathways) with complex Earth System Models (ESMs). High costs are involved in translating emission and land-use scenarios produced by Integrated Assessment Models (IAMs) into inputs for ESMs. Running these experiments on super-computers is also very expensive, and considerable labor costs are involved in setting them up, launching them and attending to their completion. Lastly, significant effort is involved in translating ESM output into datasets that can be used in impact analysis, for example through statistically downscaling and bias-correcting it.

The latest phase of the Coupled Model Intercomparison Project, Phase 6, CMIP6 (Eyring et al., 2016) prescribed standardized experiments that a large international community of modeling centers performed in order to answer a wide range of scientific questions. CMIP6 used a decentralized structure composed of self-organized MIPs, among which ScenarioMIP coordinated future scenario projections. Its experimental design (O'Neill et al., 2016) had to negotiate the trade-off between ensuring that the impact, adaptation and vulnerability (IAV) research community obtained future scenarios of relevance to their analysis framework and respecting the competing demands on ESMs' time and resources that the larger CMIP6 effort posed. Despite the latter, the modeling community signed up almost unanimously for the ScenarioMIP request – at a minimum, running the four scenarios in its Tier 1. Each experiment involved a complex set of forcing inputs (e.g., greenhouse gases and other atmospheric element concentrations, land use change trajectories) harmonized to corresponding historical estimates and downscaled from the aggregated trajectories produced by the IAMs (Gidden et al., 2019; Hurtt et al., 2020; Meinshausen et al., 2020). The computation, preparation and provision of these forcings required a complementary community effort (https://esgf-node.llnl.gov/projects/input4mips/). Outcomes from ScenarioMIP experiments form the basis for myriads of studies of the physical climate system, starting from basic characterizations of scenarios ranges and differences (Tebaldi et al., 2021) to complex and focused process-based analyses. Importantly, the same results are being used, often within the



Shared Socioeconomic Pathways-Representative Concentration Pathways (SSPs-RCPs) framework (van Vuuren et al., 2014),

to conduct integrated IAV analyses. Often, before becoming input to impact models, the ESM projections need to be downscaled and bias corrected, as done for example through the ISIMIP protocols (Lange, 2019).

The range of radiative forcing at 2100 covered by the experiments in Tier 1 of ScenarioMIP, when complemented by the Paris-inspired low warming scenario reaching only $1.9\mathrm{Wm}^{-2}$ by 2100, can be considered exhaustive of the range of future plausible outcomes, reaching up to $8.5\mathrm{Wm}^{-2}$. Ideally, however, impact analyses should be able to use an arbitrary set of sce-

narios within this range, not just the handful run by ESMs. This freedom from specific CMIP6 experiments is particularly relevant when impact analyses are conducted within an IAM framework, i.e., when the integrated assessment model endogenously produces its own trajectory of emissions and therefore global temperature changes, which should be translated into the consistent climate variables driving impacts within the same integrated modeling ecosystem. Another desirable aspect for impact risk assessment, one that also imposes a trade-off on resources, is the availability of initial condition ensembles (some-

times simply called "large ensembles") under each scenario, in order to explore the contribution of internal variability to future changes and their impacts (Lehner et al., 2020).

Thus far, the need for additional scenarios not available in ESM output archives has been addressed – when at all – by simple emulators, usually producing multi-decadal averages of temperature and – separately – precipitation change fields. Most popular has been simple pattern scaling, starting from its initial conception (Santer et al., 1990), popularized by the software

MAGICC-SCENGEN (http://www.magicc.org/, Meinshausen et al. (2011)), and made more sophisticated by the possibility of producing higher frequency fields, thus representing internal variability, for example by Link et al. (2019) and Beusch et al. (2020, 2021). More complex emulators have also been proposed departing from pattern scaling (Castruccio et al., 2014), or extensions of pattern scaling that use zonal averages to drive the emulation (Schlosser et al., 2013), or that emulate other metrics besides average temperature and precipitation (Huntingford and Cox, 2000), even extremes (Tebaldi et al., 2020). In all cases,

however, shortcomings are encountered for many applications. Impact models have evolved so that pattern scaling no longer satisfies their needs. They now often require coherent multivariate input (i.e., multiple variables that preserve their spatial and temporal correlations) and at high temporal frequencies (annual or monthly, when not higher), often spanning multidecadal periods, not just time slices. It is difficult to imagine any emulator, short of having the same complexity of an ESM, able to satisfy these requirements exhaustively.

Our approach, STITCHES, emulates ESMs by using their own output as building blocks, thus reproducing by construction the high-dimensionality, complexity and multiple frequencies of original ESM output. Working with existing scenario experiments run by an individual ESM, we stitch together time-sampled windows that we extract from the available trajectories and connect to one another in such a way that the stitched-together global temperature time series closely reproduces the global temperature time series of a new target scenario that we want to emulate. Our method can also be used to enrich existing initial

condition ensembles by adding additional synthetic members.

The idea of using existing simulations over a window when global average temperature reaches a given warming level of interest, often called time-sampling, has been frequently and prominently used in recent years (King et al., 2018; James et al., 2017). In fact, it constitutes the foundation of an entire special assessment report of the IPCC, SR1.5 (Masson-Delmotte et al.,





2018). That report's impact chapter made extensive use of this approach in the absence – at the time of its writing – of ESM

experiments that simulated low warming scenarios consistent with the Paris targets of 1.5°C or 2°C. Here we extend this

approach, which only produced isolated time windows, to the construction of entire transient scenarios. We construct climate

outcomes for the entire 21$^{st}$ century by stitching together windows of existing simulations whose global temperature trajectory

matches the scenario 'transient' warming levels, identified at regular intervals. We will show that the matching criteria we

impose are sufficient to create synthetic trajectories that are for most purposes acceptable surrogates of coherent ESM output.

In other words, we show that the stitching in most cases does not introduce significant discontinuities at the seams, or otherwise

spurious behavior for most application we can envision. The strength of our approach is that we are in practice providing a

look-up table that connects an arbitrary time series of global temperature change (the target scenario we want to emulate) to a

sequence of time windows from existing ESM simulations, from which we can extract not only global average temperature, but

importantly, complete ESM output, at any original archived frequency, in all its coherent complexity (e.g., multiple variables,

at daily frequency, on their native grid).

In the next sections, we first describe our method in detail (Sect. 2), then present results of the emulator and document the

ability of the method to reproduce the two intermediate scenarios of ScenarioMIP Tier 1 (SSP2-4.5 and SSP3-7.0) given only

the two bracketing scenarios, SSP1-2.6 and SSP5-8.5 for many of the ESMs that contributed to ScenarioMIP (Sect. 3.1.1).

We also show how the method can be used to form additional initial condition ensemble members on the basis of the existing

simulations (Sect. 3.1.2). In closing (Sect. 4), we summarize the strengths and value of our proposed emulation and discuss

its limitations, highlighting what needs to be considered before applying STITCHES in place of true ESM output. We also

propose that modeling centers could maximize their investment of resources by choosing a limited number of scenarios but

running initial condition ensembles, rather than many scenarios, suggesting that the next ScenarioMIP design may be different

from the current one.

## 2  Methods


We here describe the emulator rationale and its main aspects, and discuss our validation approach.

Many applications have in the recent past focused on a window, along the length of an ESM simulation, when global average

temperature change conforms to a given criterion (e.g., is on average 1.5°C with respect to a pre-industrial baseline). Climate

in this window is taken to be representative of conditions at that global temperature, no matter the scenario under which the

condition occurs, or the time in the simulation when the window falls. This "path-independence" assumption is valid for most

atmospheric variables, which have a short memory and whose behavior depends on the instantaneous warming level. However,

any quantity that is defined as an integral over time, like severe mega-droughts, or behaves in a way that is related to such

integral, like sea-level change, cannot be accurately represented by this method. These caveats should not be overlooked, but

for many aspects of the climate system that can be well represented by time-sampling, this approach has obviated the need for

scenarios stabilizing at low warming levels (Masson-Delmotte et al. (2018)), or has been instrumental for presenting climate

outcomes at a range of discrete warming levels, even as recently as the AR6 WG1 report of the IPCC which used global



warming levels as an alternative to scenarios for one of the dimension along which its projections are organized (Chen et al. (2021); Lee et al. (2021); Seneviratne et al. (2021); Gutiérrez et al. (2021).

Our method, that we suggestively call STITCHES, extends the time-sampling approach to an entire century-long global
average temperature trajectory, rather than just individual and discrete global average temperature targets. Our hypothesis is
that if stringent enough criteria are applied in identifying matching windows between a target global surface air tempera-
ture (GSAT) trajectory, and GSAT trajectories available in an archive of ESM simulations (e.g., through the CMIP6 database
`https://esgf-node.llnl.gov/projects/esgf-llnl/`, or the CLIVAR SMILES collection
`https://www.cesm.ucar.edu/projects/community-projects/MMLEA/`, etc.) stitching together windows will
not introduce discontinuities of consequence for most application in impact research, especially in the context of the uncertain-
ties that climate model structure or impact modeling are well known to introduce.

Our algorithm is applied separately to individual ESMs, as stitching together different models' segments would almost
certainly introduce spurious behavior. Within a single ESM universe, we can envision two distinct types of application of our
algorithm, both of which would build from existing simulations of future scenarios by that model. In one case, the goal is
to minimize the number of scenarios run by that ESM. To demonstrate the utility of STITCHES in this case, we will show
the effectiveness of the method in emulating intermediate scenarios to existing ones. This application would translate both
to an enrichment of the scenario choices for impact research, and a savings of resources for ESM simulations, particularly
when forcing inputs need to be prepared and set up. We note here, however, that by construction our algorithm does not allow
extrapolating to levels of warming above those of the highest scenario available in the archive. In another case, the goal is
to enrich the number of ensemble members available for existing scenarios. To this effect, STITCHES can be deployed on
available simulations of the target scenario and neighboring scenarios, all potential sources of usable time samples.

We now describe the steps of our algorithm.

1. GSAT from all available simulations of the 21$^{\text{st}}$ century by a given model (all scenarios and initial condition ensemble
   members) is computed; the time series is made into anomalies with respect to the average during the baseline period of
150       1995-2014 (we refer to GSAT time series in the following for brevity, but in all cases what we mean is the time series of
   GSAT anomalies);

2. a $X$-year running mean is applied to the GSAT time series and "pieces" are separated at a regular interval of the same size
   (we use $X = 9$ in our demonstration). We label these pieces derived from the existing ESM simulations as "available".
   They provide the potential building blocks to our stitching procedure;

3. for each available piece $i$ we compute its mean value, $T_i$ and, as a measure of the variation of temperature within the
   piece, the linear trend within the piece, $dT_i$;

4. the same piecing procedure is applied to the trajectory of GSAT for the target scenario; we call the result "target pieces".
   Note that in the examples we use in this paper, we derive the target GSAT trajectory from the same ESM, run under a
   scenario that we choose as target of the emulation. Therefore, we apply the smoothing procedure to the target GSAT time





series as well. Often the real application of the algorithm will target a time series of GSAT that is produced by a simple
        model, like MAGICC (http://www.magicc.org/, Meinshausen et al. (2011)) or Hector (Hartin et al., 2015), not affected
        by internal variability and therefore in no need to be smoothed. The window $X$ may be adjusted to represent the simple
        model smooth time-series more accurately by narrowing or extending $X$ until the muted year-to-year variability of the
        smooth target series is closely matched;

5. for each target piece we find all the neighbors among the available pieces, using Euclidean distance $(d_{l2})$ in the space
        $(T, X * dT)$, within a tolerance $Z$, used to define a heterogeneous matching neighborhood around each target point in
        $(T_i, X * dT_i)$ space of radius $r_i = d_{l2}\big((T_i, X * dT_i), nearest\,neighbor\big) + Z$. The choice of $Z$ could be tailored to the
        characteristics of each ESM/scenario considered, but, importantly, is also directly relevant for the number of matches
        found and therefore the number of emulated scenarios constructed.

6. one of the available pieces identified as matches/neighbors for each of the target pieces in the sequence spanning the
        21st century is chosen, and the sequence of chosen available pieces is stitched together sequentially. We can randomize
        the choice of matches, or choose nearest neighbors; importantly we do not choose the same piece more than once along
        the same emulated trajectory (one available piece may be neighbor of more than one target piece along the same target
        scenario) to avoid unrealistic repetitions, and we do not choose the same piece for the same window in time when
constructing more than one ensemble member for the same scenario, to avoid what we call "collapsed" ensembles, i.e.
        trajectories that pass through the same values year after year over a window of time. All these choices could of course
        be relaxed.

     7. once the matching/stitching steps performed over the smoothed GSAT time series are completed, the algorithm delivers
        in essence a series of pointers to the time-windows and specific experiments in the archived output from which the
pieces were extracted. Any output from the model (any variable, in isolation or jointly, at any archived frequency) can
        be stitched together according to this sequence, recreating the climate outcome of the desired variable(s) for the new
        scenario, or for an additional, synthetic member of an initial condition ensemble.

As pointed out in the description of the algorithm, its parameters $(X, Z)$ are subject to tuning. However, they both have an
interpretable function, and only small variations should be acceptable as alternative setting. In particular,

– $X$ is the size of the smoothing window and the length of the pieces used as building blocks of the synthetic time series.
        Producing a time series from the available ESM GSAT series whose smoothness matches that of the target series should
        be the aim of this parameter, when the target series does not contain internal variability. The use of about 9 years for
        GSAT has shown in our tests to be a good first guess, but trial and error for the specific simple model used may be in
        order. When starting from a time series that contains internal variability as our target, the same rule of thumb can be
a starting point for the application of the smoothing to both target and available GSAT trajectories. Naturally the time
        window also dictates the size of the piece: if the window is long enough to erase internal variability, it will prevent any
        piece from reflecting a single phase of a mode of variability, and therefore, when stitched together, the pieces will not



systematically give rise to unrealistic sequences of the phases of those modes. (This may of course fail for the longest
multi-decadal modes, like NAO or PDO, so if the impact analysis is focusing on the sensitivity of the analyzed system

to the phases of these modes, STITCHES will not be an option.)

–  $Z$ is the tolerance radius within which we identify neighbors in the two-dimensional space $(T, X * dT)$. The distance
along each dimension is immediately interpretable and comparable to the magnitude of the yearly values of GSAT within
a piece of the smoothed series (in the $T$ direction) and to the size of its variation within the piece, i.e., a measure of the
rate of temperature change within the piece (in the $X * dT$ direction) providing guidance in choosing the size of the

tolerance. The specific application may allow relaxing the criterion if a "jump" between pieces is not a concern for the
application envisioned, a beneficial choice in enlarging the number of synthetic series that can be constructed from a
finite archive of "building blocks". We also note here that fixing this tolerance in the space of the *smoothed* time series
leaves open the possibility that the original, i.e., *non-smoothed*, yearly values can present the occasional large "jump" at
the seams where the stitching is performed. We will show that this happens only occasionally, so, unless the application

is particularly sensitive to year-to-year variations, it might not be considered a fatal defect.

At the time of writing, STITCHES is built to integrate with (and depends on) the PANGEO CMIP6 archive of results
`http://gallery.pangeo.io/repos/pangeo-gallery/cmip6/`. From available runs on PANGEO, we have se-
lected all models, all experiments and all ensemble members with reported monthly gridded data for surface air temperature
and precipitation, (we will consider a smaller subset that also provides monthly gridded sea level pressure for one particular

validation exercise). Model-specific archives are created separately for each ESM. Figure 1 plots the model-specific archive of
$(T, X * dT)$ for six ESMs with various size ensembles for each of the scenarios (see 1). In the following, we either use a portion
of the archive to emulate a left-out portion of the simulations, or we use all the archive to add new "ensemble members" to
some scenarios. The former set-up simulates the situation where non-existing scenarios (those left out of the available archive)
are created from existing ones and then validated against their true realizations (which we will call "targets"). When the goal

instead is to enrich the ensemble size of existing scenarios, one has the option of using also the members of an existing initial
condition ensemble, thus producing trajectories that repeat existing ensemble members' windows, but in a difference sequence,
and mixed with other scenarios' windows. PANGEO contains files where both the historical period and the future have been
connected under the label of a specific SSP. Our emulation applies to the entire period (1850-2100), but for brevity in most of
the following we will label the various cases simply under the corresponding SSP. In fact, most of the ESMs have branched

different scenarios from the same historical simulations, so a strict out of sample construction of the historical period is in most
cases impossible. STITCHES main purpose remains the construction of future scenarios, though, so we do not worry about
this detail as we do not predicate our assessment of performance on the historical period.





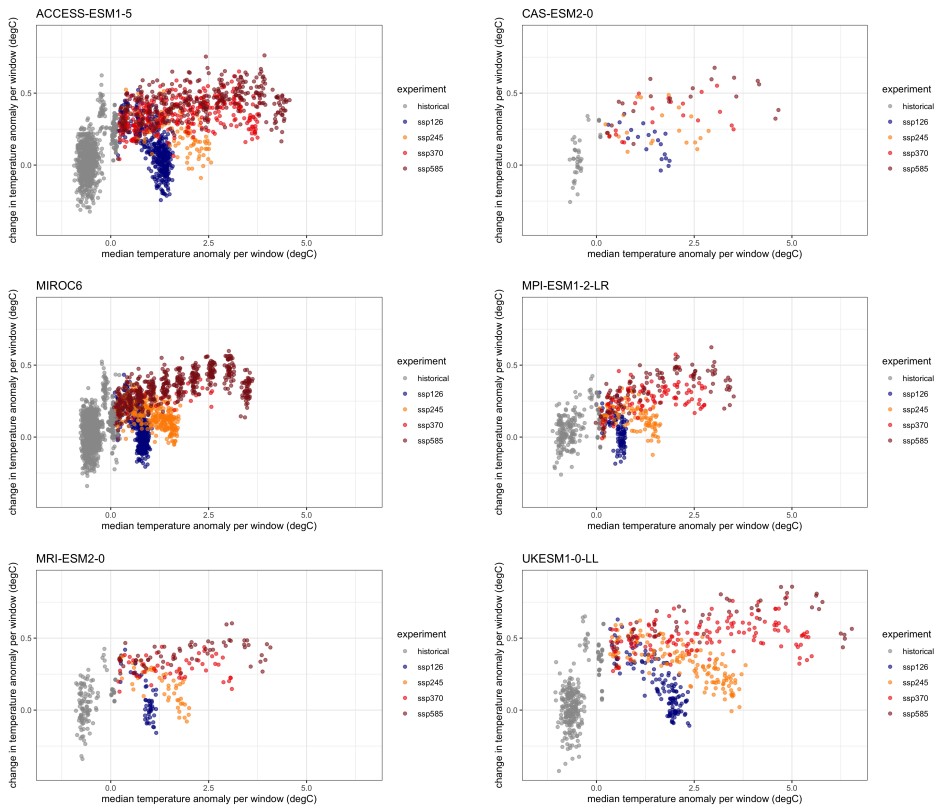

**Figure 1.** The archive content, plotted in the space of $(T, X * dT)$, i.e., the warming level with respect to the period 1995-2014 and the within window warming approximated by a linear trend, for six of the ESMs used in our emulation exercises. Each point corresponds to a $X = 9$-year-long window from an existing scenario simulation.

## 3 Results

### 3.1 General tests and validation of the synthetic series

We now show results for several test cases. Table 1 details the models, experiments and ensemble sizes from the CMIP6 archive available through the PANGEO interface as of March 15, 2022.



**Table 1.** The ESMs, experiments from ScenarioMIP O'Neill et al. (2016) and ensemble sizes from the PANGEO archive (as of 03/15/2022) used to derive test cases for our emulator.

| Model | SSP1-2.6 | SSP2-4.5 | SSP3-7.0 | SSP5-8.5 |
|---|---|---|---|---|
| ACCESS-CM2 | 5 | 5 | 5 | 5 |
| ACCESS-ESM1-5 | 40 | 10 | 30 | 35 |
| BCC-CSM2-MR | 1 | 1 | 1 | 1 |
| CAMS-CSM1-0 | 2 | 2 | 2 | 2 |
| CanESM5 | 25 | 25 | 25 | 25 |
| CAS-ESM2-0 | 2 | 2 | 2 | 2 |
| CMCC-CM2-SR5 | 1 | 1 | 1 | 1 |
| CMCC-ESM2 | 1 | 1 | 1 | 1 |
| FGOALS-g3 | 4 | 4 | 4 | 4 |
| FIO-ESM-2-0 | 3 | 3 | 0 | 3 |
| GISS-E2-1-G | 4 | 5 | 9 | 4 |
| HadGEM3-GC31-LL | 1 | 4 | 0 | 4 |
| MCM-UA-1-0 | 1 | 1 | 1 | 1 |
| MIROC-ES2L | 10 | 30 | 10 | 10 |
| MIROC6 | 50 | 50 | 3 | 50 |
| MPI-ESM1-2-HR | 2 | 2 | 10 | 2 |
| MPI-ESM1-2-LR | 10 | 10 | 10 | 10 |
| MRI-ESM2-0 | 5 | 5 | 5 | 6 |
| NESM3 | 2 | 2 | 0 | 2 |
| NorESM2-LM | 1 | 3 | 3 | 1 |
| NorESM2-MM | 1 | 2 | 1 | 1 |
| TaiESM1 | 1 | 1 | 1 | 1 |
| UKESM1-0-LL | 13 | 14 | 13 | 5 |

Note that when the goal is emulating non-existing scenarios, our targets need to be trajectories that reach warming levels lower than or equal to the ones available as building blocks in the archive, as our algorithm does not allow extrapolating. Similarly, STITCHES stops short of being able to emulate overshoot scenarios, given that the archive does not offer a large
population of overshoot experiments that we can use as building blocks (i.e., the cooling behavior of GSAT in an overshoot experiment cannot be sampled from increasing, or flat, GSAT trajectories). These considerations could be useful to keep in mind when designing the next phase of ScenarioMIP.





### 3.1.1 Validation of emulated intermediate scenarios

**Table 2.** The number of emulated trajectories produced to assess the performance of STITCHES in recreating intermediate scenarios (SSP2-4.5 and SSP3-7.0) from the two "bracketing" scenarios (SSP1-2.6 and SSP5-8.5).

| Model | SSP2-4.5 | SSP3-7.0 |
|---|---|---|
| ACCESS-CM2 | 4 | 3 |
| ACCESS-ESM1-5 | 5 | 5 |
| BCC-CSM2-MR | 1 | 1 |
| CAMS-CSM1-0 | 2 | 2 |
| CanESM5 | 3 | 1 |
| CAS-ESM2-0 | 2 | 1 |
| CMCC-CM2-SR5 | 1 | 1 |
| CMCC-ESM2 | 1 | 1 |
| FGOALS-g3 | 4 | 4 |
| FIO-ESM-2-0 | 3 | - |
| GISS-E2-1-G | 4 | 5 |
| HadGEM3-GC31-LL | 1 | - |
| MCM-UA-1-0 | 1 | 1 |
| MIROC-ES2L | 3 | 5 |
| MIROC6 | - | 3 |
| MPI-ESM1-2-HR | 2 | 2 |
| MPI-ESM1-2-LR | 5 | 4 |
| MRI-ESM2-0 | 1 | 3 |
| NESM3 | 2 | - |
| NorESM2-LM | 1 | 1 |
| NorESM2-MM | 1 | 1 |
| TaiESM1 | 1 | 1 |
| UKESM1-0-LL | 3 | 4 |

Our first goal is to test the ability of STITCHES to reconstruct new scenarios from existing scenarios. We do so for all

available ESMs in the PANGEO CMIP6 archive that provide at least one member under SSP1-2.6 and one member under SSP5-8.5, targeting the two intermediate scenarios SSP2-4.5 and SSP3-7.0 (See Table 1). Intentionally, we set the two parameters $(X, Z)$ to the same values, independently of the specific ESM targeted. A specific choice could only ameliorate the performance of our emulator for any given model used as test case. However, our common choice is the result of considering the behavior of many ESMs and finding values that are consistent with most, so, de facto, these parameters are tailored to some ESMs and less



tailored to others. The best performance that we document could be regarded as what is expected when tailoring the parameter values to the specific ESM that we want to emulate.

Table 2 lists the number of emulated trajectories for each of the intermediate scenarios tested, and for each of the models. Since this exercise is not about producing many replicates, but simply reproducing a target trajectory, for each model we set out to reproduce as many targets trajectory as there are ensemble members available under SSP2-4.5 or SSP3-7.0 if such number is

less than or equal to five, capping at five the number of targets also for those models with more ensemble members potentially available as targets (as Table 1 reports).

As mentioned in Sect. 2 our emulation approach produces the same complex, multidimensional output as an ESM does. Thus, validation could take an infinite number of forms, over a range of variables in isolation or jointly, and over arbitrary space and time scales. To simplify the task, however, we rely here on the well known result that – among atmospheric variables that are commonly used for impact modeling – surface air temperature has, relatively speaking, a lower amount of internal

variability, and this variability becomes lower the larger the averages taken in the time and space domains. Thus, our validation will start by considering the behavior of *annual average* GSAT trajectories from STITCHES.

Our first concern is to not *systematically* introduce significant discontinuities by stitching together separate windows of ESM output, often from altogether different experiments (if always from the same ESM). To this end, we consider the year-to-year

difference in the annual GSAT trajectories stitched together. The tolerance allowed for the match ($Z = 0.075$) is responsible for keeping the stitched-together pieces of the *smoothed trajectories* within a narrow interval of one another, but cannot directly control what happens when we recover the stitched-together original trajectories of (non-smoothed) annual values for this validation exercise. These could differ by a larger amount if, by chance, the 9-year pieces happen to end/begin with widely different values. Our concern is that this do not happen systematically. Table 3 reports, for each ESM, how many of these seams

(as many for each trajectory as there are 9-year intervals) produce a year-to-year variation that is larger than twice the standard deviation of the real year-to-year variations. The latter are taken as either those from the archive simulations used as building blocks within the stitched trajectories (thus addressing the question "do the seams stand out from the rest of the series within which they appear?") or those in the target series (thus addressing the question "do the seams stand out compared to the year-to-year variations of the trajectories we want to emulate?"). As can be assessed, this behavior emerges only very sporadically,

with most cases well below 10% of the seams. In fact the mean of these values is just above 5%, which could be the expected outcome by chance of such an exercise, even if those outliers came from the same distributions used as comparison.

We then compare linear trends fitted to the stitched trajectories to linear trends fitted to the target series, by separately fitting a linear trend to the historical period (1850-2014) and the future period, 2015-2100. The trends are defined as the angular coefficient of a linear regression of annual mean values of GSAT onto years, and we consider central estimates (by

ordinary least squares) and 95% confidence intervals. We find that in all cases (109 stitched trajectories across the models and the two scenarios) historical trend central estimates for the stitched series fall comfortably within the confidence intervals of the historical trends of the target series. For the future trends, the confidence intervals of the stitched series overlap with the confidence intervals of the trends from the target series in all cases. There are 21 trajectories out of the 109 for which the central estimates fall outside those confidence intervals. In all these cases, the difference between the central estimate and the closest





**Table 3.** For all the models used in our emulation of SSP2-4.5 and SSP3-7.0 we report the number of "seams" at which annual GSAT presents a jump that is larger than twice the inter-annual standard deviation. The latter is computed either from the interannual variations of the archive simulations used in the stitching (in practice, the interannual standard deviations of the stitched trajectories without including the seams in its computation), or from the target experiments (the interannual standard deviations of the real series that we are emulating). We also show the total number of seams from which the percentages mentioned in the text are computed.

| Model | SSP2-4.5 | | | SSP3-7.0 | | |
|---|---|---|---|---|---|---|
| | fraction (vs. archive) | fraction (vs. target) | total seams | fraction (vs. archive) | fraction (vs. target) | total seams |
| ACCESS-CM2 | 9 | 11 | 108 | 4 | 6 | 81 |
| ACCESS-ESM1-5 | 1 | 1 | 135 | 9 | 10 | 270 |
| BCC-CSM2-MR | 2 | 2 | 27 | 2 | 2 | 27 |
| CAMS-CSM1-0 | 3 | 3 | 54 | 3 | 3 | 54 |
| CanESM5 | 2 | 3 | 81 | 0 | 0 | 27 |
| CAS-ESM2-0 | 0 | 0 | 54 | 2 | 2 | 27 |
| CMCC-CM2-SR5 | 0 | 0 | 27 | 3 | 2 | 27 |
| CMCC-ESM2 | 1 | 1 | 27 | 1 | 1 | 27 |
| FGOALS-g3 | 2 | 5 | 108 | 5 | 5 | 108 |
| FIO-ESM-2-0 | 1 | 3 | 81 | | | |
| GISS-E2-1-G | 1 | 2 | 108 | 4 | 4 | 135 |
| HadGEM3-GC31-LL | 1 | 1 | 27 | | | |
| MCM-UA-1-0 | 3 | 2 | 27 | 4 | 4 | 27 |
| MIROC-ES2L | 3 | 3 | 81 | 3 | 4 | 135 |
| MIROC6 | 1 | 1 | 108 | 1 | 1 | 81 |
| MPI-ESM1-2-HR | 2 | 3 | 54 | 2 | 4 | 54 |
| MPI-ESM1-2-LR | 5 | 6 | 135 | 3 | 3 | 108 |
| MRI-ESM2-0 | 3 | 1 | 27 | 5 | 5 | 81 |
| NESM3 | 3 | 3 | 54 | | | |
| NorESM2-LM | 3 | 3 | 27 | 1 | 1 | 27 |
| NorESM2-MM | 3 | 4 | 27 | 2 | 2 | 27 |
| TaiESM1 | 2 | 2 | 27 | 3 | 3 | 27 |
| UKESM1-0-LL | 3 | 3 | 81 | 4 | 5 | 108 |

bound of the confidence interval is a very small value: in one single case, the central estimate is outside the confidence intervals by 0.056°C per decade. In two more cases, the values are between 0.04°C and 0.05°C; six more cases miss by 0.01-0.023°C per decade, with the remaining 12 cases falling outside the respective confidence intervals only by 0.01°C per decade or less.



We also compute inter-annual standard deviations for target and stitched trajectories, finding that once again, historical simulations remain within the ranges of the target trajectories in all cases. For the future period, in 78% of cases, the stitched series show inter-annual variability within 20% of that of the target series. The remaining 24 cases out of the 109 tested whose interannual variations fall outside the range of the target series show discrepancies that amount to less than 0.2°C in value in all cases, with a median value of 0.004°C and a third quartile of 0.05°C. Last, we compute autocorrelation and partial autocorrelation to determine the frequency characteristics of the time series. The results confirm the similarities of stitched and target series, i.e, the emulated trajectories do not show spurious behavior, with discrepancies in the AR order estimated only in those cases when the higher orders are at the margin of statistical significance (not shown).

Even if for a large majority of cases the performance of the emulator seems acceptable, and in many cases indistinguishable from the target cases, we underline that some model/experiment combinations appear to be challenging for this uniform set up. Most of these cases coincide with models providing only one ensemble member per scenario, and the spurious behavior is often found at the higher end of the warming range within the scenario emulated, where the only possible matches come from the model's only SSP5-8.5 available trajectory. It is not unlikely that the matches from the higher scenario result in less than optimal windows, given the limited choice available for the higher temperature levels. Likely, fixing the tolerance parameter to a tighter value could improve these specific emulation cases, or simply fail to create an emulated trajectory, so that the user would have an outright warning of the difficulty in matching. Here we remain within a generic setting in order to show the trade-offs at play, and identify lessons. We show in Figure 2 through Figure 4 some examples of target (in black) and stitched (colored lines) GSAT trajectories for the two intermediate scenarios and many of the models we test. Also from these figures one can assess that the behaviors that appear to deviate from the expected, are all at the tail end of the simulations, and only for those models that offer only one pair of scenarios in the archive to sample from. In these figures we use a range of colors, from cool to warm hues, to give a sense of the number of trajectories plotted in these spaghetti diagrams: while the target ensemble is always drawn in black lines, the emulated trajectories are in color, with cases showing warmer colors being those where we have created a larger number of stitched trajectories.



**Figure 2.** Examples of target (black lines) and stitched (colored) GSAT time series for ESMs in the PANGEO archive that ran at least one trajectory along the Tier 1 experiments of ScenarioMIP (SSP1-2.6; SSP2-4.5; SSP3-7.0; SSP5-8.5). We use the two bracketing scenarios to stitched together trajectories that follow the two intermediate scenarios, and compare true and stitched. As CMIP6 experiments were designed to branch multiple scenarios from the same historical simulation, when the model provides only few ensemble members in the archive chances are that some (at times, many) of the historical windows find the closest matches to be "themselves", only labelled SSP1-2.6 or SSP5-8.5. That explains the perfect match in the historical periods between target and stitched trajectories. We do not worry about these cases, as the focus of our exercise is the creation of future scenarios.





**Figure 3.** Examples of target (black lines) and stitched (colored) GSAT time series for ESMs in the PANGEO archive that ran at least one trajectory along the Tier 1 experiments of ScenarioMIP (SSP1-2.6; SSP2-4.5; SSP3-7.0; SSP5-8.5). We use the two bracketing scenarios to stitched together trajectories that follow the two intermediate scenarios, and compare true and stitched. As CMIP6 experiments were designed to branch multiple scenarios from the same historical simulation, when the model provides only few ensemble members in the archive chances are that some (at times, many) of the historical windows find the closest matches to be "themselves", only labelled SSP1-2.6 or SSP5-8.5. That explains the perfect match in the historical periods between target and stitched trajectories. We do not worry about these cases, as the focus of our exercise is the creation of future scenarios.



**Figure 4.** Examples of target (black lines) and stitched (colored) GSAT time series for ESMs in the PANGEO archive that ran at least one trajectory along the Tier 1 experiments of ScenarioMIP (SSP1-2.6; SSP2-4.5; SSP3-7.0; SSP5-8.5). We use the two bracketing scenarios to stitched together trajectories that follow the two intermediate scenarios, and compare true and stitched. As CMIP6 experiments were designed to branch multiple scenarios from the same historical simulation, when the model provides only few ensemble members in the archive chances are that some (at times, many) of the historical windows find the closest matches to be "themselves", only labelled SSP1-2.6 or SSP5-8.5. That explains the perfect match in the historical periods between target and stitched trajectories. We do not worry about these cases, as the focus of our exercise is the creation of future scenarios.





For all cases when the emulation of GSAT does not present inconsistencies for annual average values, our hypothesis is that noisier quantities would not suffer from detectable discontinuities either. We have tested this expectation for a range of quantities (temperature, precipitation, and sea level pressure) and scales (from subcontinental to local, i.e. grid-point level) confirming it. Here we compare trends and variability (computed as the standard deviations of the residuals from the trend)

between stitched and target time series under the two scenarios (over the 2015-2100 period) for temperature (TAS) and precipitation (PR) at the grid point level using monthly time series. We use results from the emulation of two models that represent extremes in the PANGEO dataset, in terms of availability of archive trajectories: CAMS-CM1-0 (with only two ensemble members each for SSP1-2.6 and SSP5-8.5) for which we have derived one emulated trajectory per scenario (SSP2-4.5 and SSP3-7.0) and MIROC6 (with 50 ensemble members for each) for which we have emulated three trajectories per scenario. In

the trend figures we blacken grid-points where the trends computed from the stitched trajectories are substantially different from those computed from the target trajectory. We use here the same criterion that we applied to the validation of GSAT: trends are significantly different when their 95% confidence intervals do not overlap. For the analysis of monthly variability we show maps of the ratio of the two variances computed from the stitched and target time series, after removing the linear trends. We consider substantially different variances that are not within 20% of one another, i.e. whose ratio is either less than 0.8 or

more than 1.2. The color bar is chosen to highlight these two thresholds. Figure 5 shows results for the comparison of TAS and PR trends for CAMS-CM1-0 while Figure B1 through B3 in the appendix show the corresponding analysis for MIROC6. For temperature, as can be assessed in Figure 5, top panels, only isolated patches over the tropical oceans show statistically significant differences in trends. The results for MIROC6, where we can look at three different realizations, show that also for this model emulation the areas of disagreement consist of isolated patches mostly over ocean regions, and not consistent

from realization to realization, suggesting that there may be some internal variability still at play influencing these results, rather than a systematic problem with STITCHES. Internal variability is likely responsible for an area in the Arctic appearing as inconsistent in two of the realizations. For precipitation the inconsistent areas are barely detectable as smaller scatters of points, mostly over the oceans.





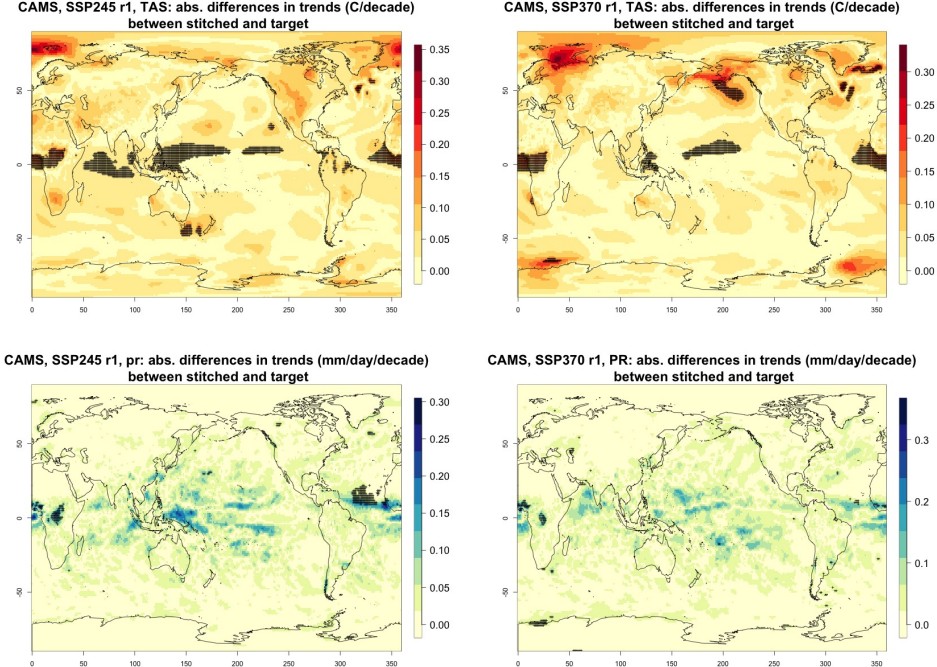

**Figure 5.** Absolute difference in decadal trends of temperature (TAS) and precipitation (PR) between stitched and target realizations. The value of the difference is expressed by the color scale and we marked as significant by black crosses those locations where the trends computed from target and stitched time series do not overlap in their 95% confidence intervals, indicating statistically significant differences. Emulation of CAMS-CM1-0 monthly time series for 2015-2100 under SSP2-4.5 and SSP3-7.0.



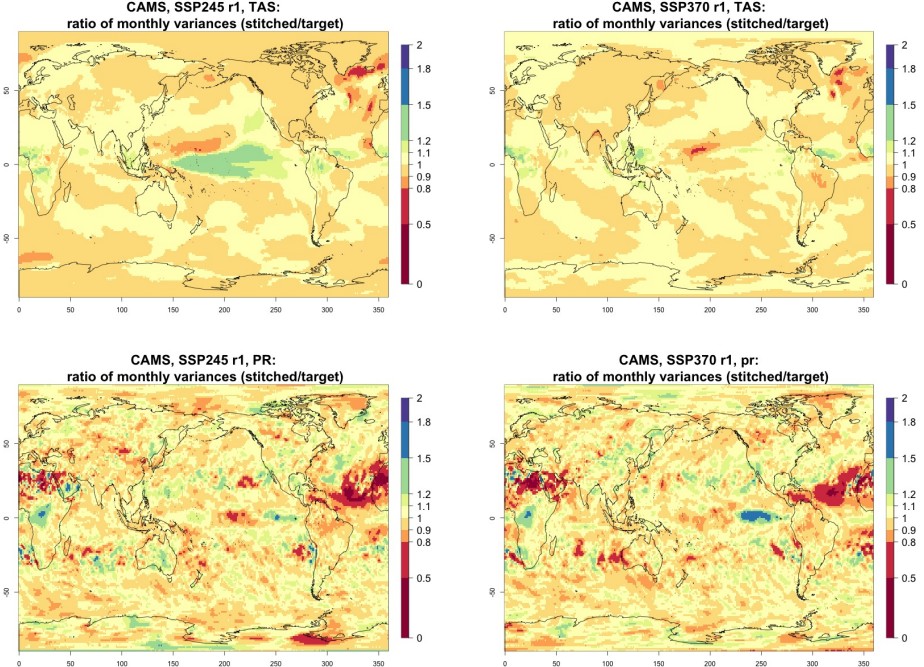

**Figure 6.** Ratio in monthly variability (standard deviation of residuals from trends) of temperature (TAS) and precipitation (PR) between stitched (at the numerator) and target (at the denominator) realizations. The value of the ratio is expressed by the color scale which highlights the transition at 0.8 and 1.2. Emulation of CAMS-CM1-0 monthly time series for 2015-2100 under SSP2-4.5 and SSP3-7.0.

Performance in terms of monthly variability of temperature is within 20% of the true variability practically over all the land

regions, and over the large majority of the oceans' areas, with the exception of a systematic bias over the west Pacific cold tongue. Rainfall variability appears less homogeneously accurate, until one realizes that the areas where variability appears inconsistent (i.e., areas where the value of the ratio is smaller than 0.8, or larger than 1.2) coincide with climatologically very dry areas, of both the Northern and Southern hemispheres. In these regions variability is low, and therefore small differences in the numerator and denominator may cause large variation of the ratio, without implying meaningful differences in rainfall

behavior.

Last, still concerned with single time series behavior, we consider a different quantity altogether: the SOI index, describing the evolution of the El-Niño Southern Oscillation mode of variability. The SOI index is defined as the standardized difference between sea level pressure (SLP) monthly anomalies at Tahiti and Darwin, Australia (see https://www.ncdc.noaa.gov/teleconnections/enso/indicators/soi/). The negative or positive sign of this difference indicates abnormally warm or cold

ocean waters across the eastern tropical Pacific, associated with El Niño or La Niña episodes. The index, despite being a uni-dimensional time series, reflects the behavior of a coherent spatial field (SLP) at a monthly frequency. Its frequency characteristics are important to preserve, as the opposite phases of the SOI produce significant shifts in the weather of regions where SOI teleconnections are strong, causing droughts or intense precipitation, cooler or warmer than average conditions. Therefore,





for impact analysis, we would not want to produce time series of this index with a spurious behavior, compared to the corre-

sponding continuous output of the emulated ESM (note however that we are not comparing these frequency characteristics to observations, which is not the point of our emulator). We consider this validation particularly important, both because of the salience of ENSO behavior for many types of impact, and because the frequency characteristics of this mode of variability are close to our 9-year windows.

Figure 7 presents target and stitched time series of the ENSO index for one of the models (CAMS-CM1-0) and three twenty

year windows along the two scenarios emulated. As can be gauged, the three pairs of time series appear similar in magnitude and oscillatory behavior. In order to confirm the latter, we show in Figure 8 the (partial) auto-correlation functions of the corresponding time series. This analysis produces indistinguishable lag patterns, and, importantly, does not reveal any spurious behavior at frequencies of 9 years, i.e., at frequencies reflecting the spacing of the seams. Figures C1 through C6 in the appendix confirm that results are similar for three emulated ensemble members under each scenario for MIROC6.





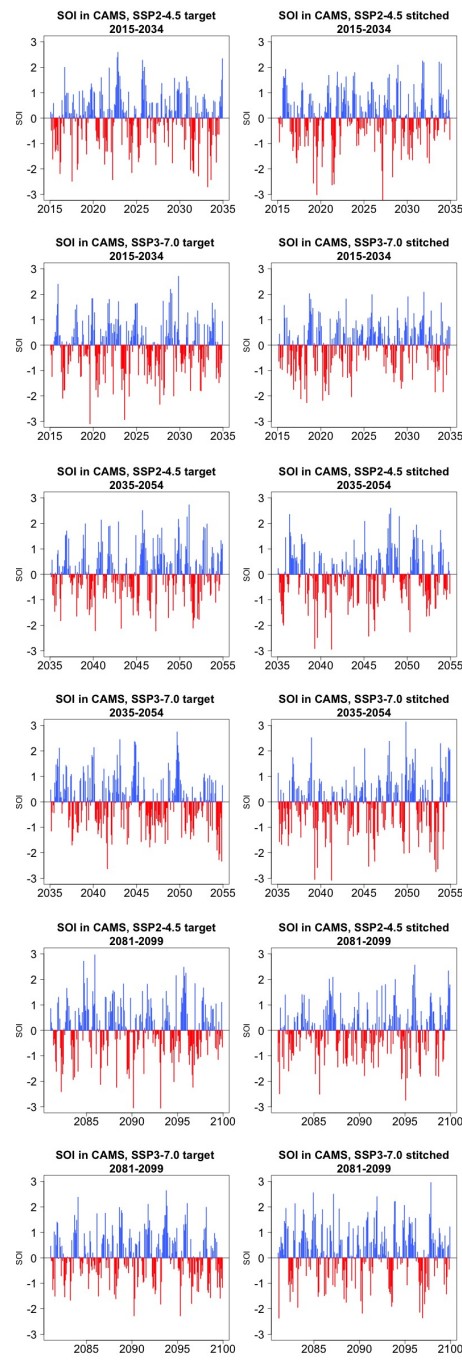

**Figure 7.** Examples of target (left) and stitched (right) SOI time series for three twenty-year windows along the length of the simulation: 2015-2034 in the top four panels; 2035-2054 in the middle four panels; 2081-2100 in the bottom four panels. Results from emulation of SSP2-4.5 and SSP3-7.0 for CAMS-CM1-0.





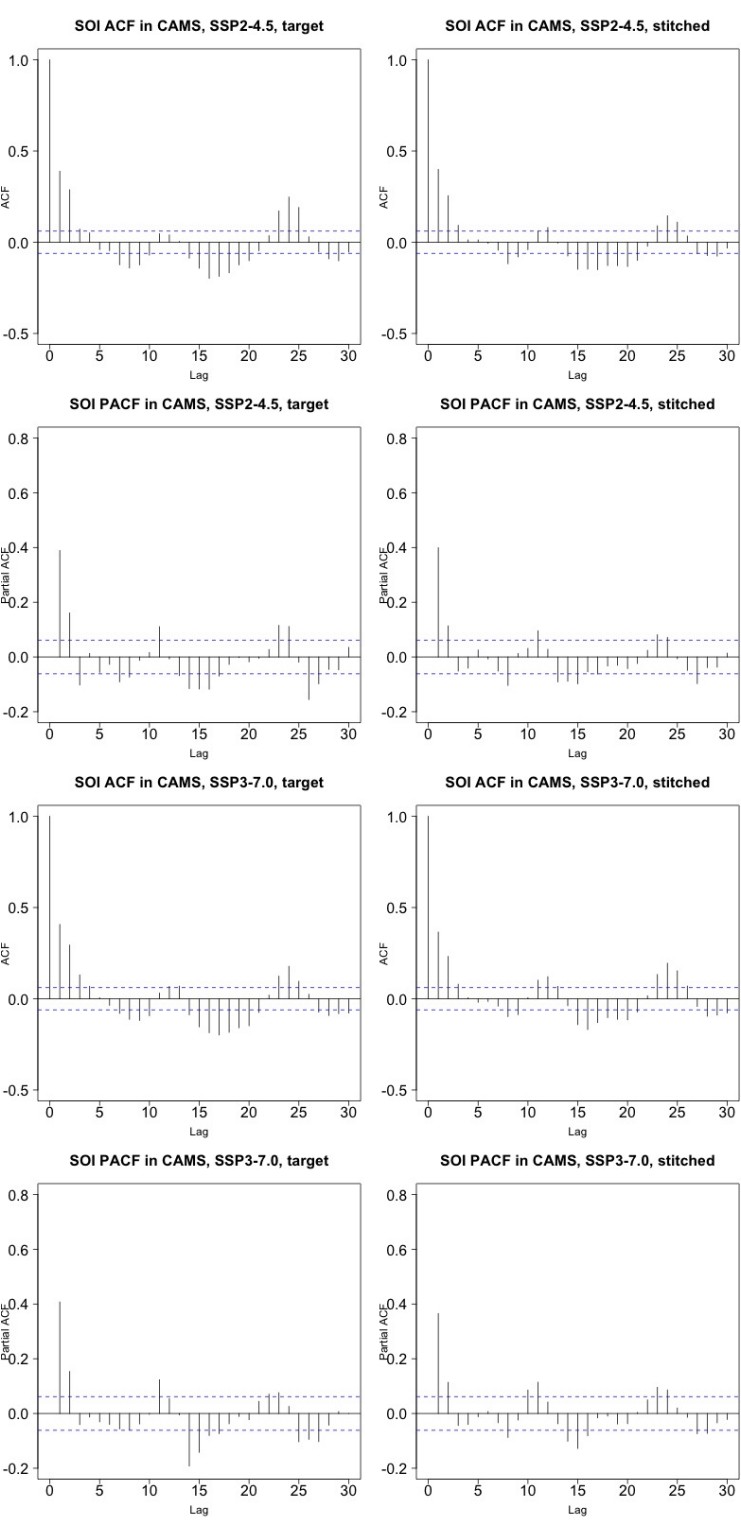

**Figure 8.** Auto-correlation (ACFs) and Partial auto-correlation functions (PACFs) for real and stitched SOI time series. Top two rows: SSP2-4.5 ACF for target and stitched series and respective PACFS. Bottom two rows: SSP3-7.0 ACF for target and stitchedseries and respective PACFs.



### 3.1.2 Validation of emulated initial condition members

Our emulator can also be used to provide multiple ensemble members under the same scenario, akin to initial condition ensembles. For this type of application, besides the necessary validation of the individual member according to the above described metrics, we want to validate the properties of the synthetic ensembles as such, comparing their mean behavior and their spread to those of real initial condition ensembles from the same ESM. Figures D1 through D5 show the resulting ensembles for a number of experiments that we conducted over several ESMs and the two scenarios SSP2-4.5 and SSP3-7.0. We chose models that provided at least 5 21$^{\text{st}}$ century trajectories of the Tier 1 scenarios. As mentioned in Section 2, this exercise is conducted by using the entire archive available, as we mimic a situation where we are not creating a new scenario, but augmenting the size of an ensemble under existing ones.

We adopt the two-dimensional metric of performance introduced by Tebaldi et al. (2020),

$$Er(\mathbf{y}, \hat{\mathbf{y}}) = \left( \frac{|\bar{\mathbf{y}} - \bar{\hat{\mathbf{y}}}|}{\sqrt{< (\mathbf{y} - \bar{\mathbf{y}})^2 >}}, \frac{\sqrt{< (\hat{\mathbf{y}} - \bar{\hat{\mathbf{y}}})^2 >}}{\sqrt{< (\hat{\mathbf{y}} - \bar{\mathbf{y}})^2 >}} \right) . \tag{1}$$

Its first component (which we indicate below as $E_1$) measures the systematic bias between the means of the synthetic and the true ensembles, here indicated by $\mathbf{y}$ and $\hat{\mathbf{y}}$ respectively, rescaled by the true ensemble variance. The second, $E_2$ is defined as the ratio between the synthetic and the true ensemble variances. Obviously, we want the former to be close to zero and the latter to be close to one for our method to emulate accurately the ESM initial condition ensemble behavior. It is useful to note that the magnitude of these metric components is expressed as a fraction (or multiple) of the true ensemble variance, allowing a judgment of the size of the discrepancies introduced by STITCHES as they compare to the true internal variability of the target ensemble. Here as before we focus on annual series of global mean temperature. In Table 4 the values of $E_1$ and $E_2$ are reported. The number under the column labelled "Archive Size" indicates how many 21$^{\text{st}}$ century trajectories were available for each of the four scenarios in the archive to create 9-year building blocks for STITCHES. Note that when a model provided numerous trajectories to build from, we tested the performance for increasing sizes of the archive (e.g., for the CanESM5 model we repeat the exercise using 5-,10-,15-,20-,25-members initial condition ensembles for each scenario). The following two columns in the tables list the size of the target ensemble emulated, which is therefore available for validation ($\mathbf{y}$), under "Target Members", and the size of the stitched ensemble, under "Stitched Members", which is the number of additional trajectories created by STITCHES that could be added to the existing ensemble. We choose three years along the 21st century, 2010, 2050 and 2090, and we utilize 9-year windows around those years to compute formula 1 (similar results were obtained by using a shorter 5-year window).

Several outcomes can be gleaned from Table 4. STITCHES trajectories have mean and variability characteristics within a small window of the target ensembles in the great majority of cases. Of course the application will dictate what is the standard that needs to be met by the synthetic ensembles, but if discrepancies of up to 25 or 30% of the true internal variability are acceptable, most cases described in the table would meet that standard, and in the great majority of cases by a large margin, especially once the ensembles available from which we resample building blocks have at least 10 members. This exercise uses a tolerance value $Z = 0.075$ across the board, but tuning the value to specific ESM characteristics could ameliorate some of





**Table 4.** The two components of the $Er$ metric, $E_1$ and $E_2$, computed for several experiments across ESMs, scenarios and number of available archive trajectories from which to create the stitched ensembles. Numbers in columns 4 through 9 represent fractions of the target ensemble variance (see formula 1).

| Model | Scenario | Archive Size | Target Members | Stitched Members | $E_1$ 2010 | $E_1$ 2050 | $E_1$ 2090 | $E_2$ 2010 | $E_2$ 2050 | $E_2$ 2090 |
|---|---|---|---|---|---|---|---|---|---|---|
| ACCESS-CM2 | ssp245 | 5 | 5 | 5 | 0.00 | 0.36 | 0.00 | 1.03 | 0.82 | 1.09 |
| ACCESS-ESM1-5 | ssp245 | 5 | 10 | 6 | 0.29 | 0.00 | 0.02 | 1.16 | 1.16 | 1.35 |
| CanESM5 | ssp245 | 5 | 25 | 5 | 0.15 | 0.13 | 0.36 | 0.68 | 1.26 | 0.87 |
| MIROC-ES2L | ssp245 | 5 | 30 | 5 | 0.08 | 0.50 | 0.07 | 0.96 | 0.95 | 1.00 |
| MPI-ESM1-2-LR | ssp245 | 5 | 10 | 5 | 0.01 | 0.03 | 0.25 | 1.07 | 1.06 | 1.26 |
| MRI-ESM2-0 | ssp245 | 5 | 5 | 4 | 0.00 | 0.19 | 0.11 | 1.09 | 1.09 | 1.21 |
| UKESM1-0-LL | ssp245 | 5 | 14 | 3 | 0.04 | 0.51 | 0.37 | 1.18 | 0.99 | 0.93 |
| ACCESS-ESM1-5 | ssp245 | 10 | 10 | 9 | 0.00 | 0.02 | 0.00 | 1.40 | 0.99 | 1.18 |
| MIROC-ES2L | ssp245 | 10 | 30 | 10 | 0.01 | 0.00 | 0.02 | 1.20 | 1.17 | 1.01 |
| MPI-ESM1-2-LR | ssp245 | 10 | 10 | 10 | 0.00 | 0.04 | 0.01 | 1.06 | 0.89 | 0.86 |
| CanESM5 | ssp245 | 10 | 25 | 10 | 0.02 | 0.01 | 0.01 | 1.02 | 1.13 | 0.91 |
| CanESM5 | ssp245 | 15 | 25 | 15 | 0.01 | 0.01 | 0.01 | 0.92 | 0.87 | 0.99 |
| CanESM5 | ssp245 | 20 | 25 | 20 | 0.01 | 0.01 | 0.02 | 1.02 | 1.02 | 0.99 |
| CanESM5 | ssp245 | 25 | 25 | 23 | 0.00 | 0.00 | 0.00 | 0.88 | 1.02 | 1.00 |
| ACCESS-CM2 | ssp370 | 5 | 5 | 5 | 0.00 | 0.00 | 0.00 | 0.89 | 0.91 | 1.00 |
| ACCESS-ESM1-5 | ssp370 | 5 | 30 | 5 | 0.05 | 0.14 | 0.02 | 1.09 | 1.04 | 1.40 |
| CanESM5 | ssp370 | 5 | 25 | 4 | 0.30 | 0.03 | 0.00 | 1.40 | 0.85 | 0.90 |
| MIROC-ES2L | ssp370 | 5 | 10 | 5 | 0.06 | 0.69 | 0.01 | 1.18 | 1.16 | 0.90 |
| MPI-ESM1-2-LR | ssp370 | 5 | 10 | 5 | 0.27 | 0.30 | 0.00 | 1.28 | 1.15 | 1.20 |
| MRI-ESM2-0 | ssp370 | 5 | 5 | 5 | 0.00 | 0.07 | 0.01 | 0.90 | 1.13 | 1.14 |
| UKESM1-0-LL | ssp370 | 5 | 13 | 3 | 0.01 | 0.00 | 0.09 | 0.94 | 1.30 | 1.03 |
| ACCESS-ESM1-5 | ssp370 | 10 | 30 | 11 | 0.00 | 0.07 | 0.01 | 1.25 | 0.93 | 0.91 |
| CanESM5 | ssp370 | 10 | 25 | 10 | 0.11 | 0.00 | 0.03 | 0.87 | 1.02 | 1.03 |
| MIROC-ES2L | ssp370 | 10 | 10 | 9 | 0.11 | 0.22 | 0.00 | 1.11 | 1.08 | 0.98 |
| MPI-ESM1-2-LR | ssp370 | 10 | 10 | 9 | 0.01 | 0.00 | 0.01 | 1.12 | 1.46 | 1.05 |
| CanESM5 | ssp370 | 15 | 25 | 14 | 0.01 | 0.07 | 0.03 | 0.94 | 0.91 | 1.02 |
| CanESM5 | ssp370 | 20 | 25 | 19 | 0.00 | 0.00 | 0.01 | 0.90 | 0.96 | 1.02 |
| CanESM5 | ssp370 | 25 | 25 | 23 | 0.01 | 0.00 | 0.00 | 0.95 | 1.01 | 1.01 |

the worse performances. A look at the best performances suggests what we would expect if this type of tuning was conducted





specifically to each ESM characteristics of variability. Section 3.1.3 below further expands on the relation between the tuning
parameter $Z$, the size of the ensembles that STITCHES can create, and the values of the $Er$ metric.

We have performed the same exercise by limiting the archive to the two bracketing scenarios, SSP1-2.6 and SSP5-8.5, and trying to construct ensembles for SSP2-4.5 and SSP3-7.0. In this case STITCHES is significantly challenged, and its performance as measured by $Er$ more often worst that the 25-30% standard. Table E1 reports these discrepancies. A look at Figure 1 may suggest the nature of the challenge here, because of the relatively extreme nature of SSP5-8.5 values compared to
SSP2-4.5 in particular. Section 3.1.3 below also discusses this aspect. We argue that this challenge could be lessened by a more deliberate design of ESM experiments in relation to the $(T, X * dT)$ space, together with the fact that, as it has been argued recently, SSP5-8.5 may represent an obsolete or at least improbable scenario Hausfather and Peters (2020) and therefore the range to be explored by future scenarios should be narrower in the next phase of CMIP experiments.



### 3.1.3 Trade-offs between generated ensemble size and $Z$

The size of a stitched ensemble targeting a given experiment is directly related to the number of ESM ensemble members present in the archive, as well as the tolerance for matching, $Z$. Larger values of $Z$ result in larger numbers of stitched ensemble members, until the archive is exhausted. It is unlikely that a closed form relationship between (archive size, $Z$) and size of the stitched ensemble exists, as another factor in the success of the emulation is how similar the GSAT trajectories in the archive are to the target and archive scenarios, not only in mean but in rate of warming, the two dimensions of our neighborhoods.

Instead, we present empirical estimates, for each ESM separately, of a conservative cutoff value for $Z$, $Z_{cutoff}$ that should safely result in generated ensemble members satisfying our validation criteria presented in the above Sections 3.1.1 and 3.1.2. Specifically, we will identify the $Z_{cutoff}$ at which the generated ensemble size appears to saturate while still maintaining small $Er$ values. Thus, using $Z$ values beyond the provided $Z_{cutoff}$ provides no additional benefit.

To identify $Z_{cutoff}$ for each experiment of each ESM, we conduct a sweep of $Z$ values ranging from 0.04 to 0.3°C. As

noted in step 6 of the algorithm, the actual matches within each $Z$ neighborhood are drawn randomly for stitching a trajectory. Therefore, at each $Z$ value tested, we perform 50 of these random draws of the STITCHES algorithm for generating the largest possible collapse-free ensemble, targeting each of experiment SSP2-4.5 and SSP3-7.0, and using the full archive for that ESM (see Table 1). We calculate the same $Er$ statistics above for both GSAT and the GSAT differences "at the seams" computed over the annual time series of stitched GSAT, for each draw of a full generated ensemble. $Z_{cutoff}$ is identified as the largest

tolerance value that keeps the average value across draws of the two pairs of these metrics for each target ensemble below 10%. Generally, it is the $E_2$ dimension of the GSAT differences at the seams when targeting SSP2-4.5 that is largest of the four error metrics across both target experiments: i.e., the standard deviation of the generated interannual jumps differs from the standard deviation of the target interannual jumps of the target.

$Z_{cutoff}$ values and the corresponding draw-averaged generated ensemble size for each experiment-ESM combination are

reported in Table 5. Increasing the tolerance beyond these $Z_{cutoff}$ can increase the generated ensemble size, but with larger errors, meaning potentially the stitched realizations at that point may not behave well. For example, at $Z_{cutoff} = 0.25$, ACCESS-CM2 can stitch seven realizations targeting SSP2-4.5 but at max error of 11.6% ($E_2$ of the GSAT differences in this case). Values of $E_2$ of the GSAT differences this large may correspond to stitched GSAT trajectories that clearly switch between SSP1-2.6 and SSP5-8.5 windows, rather than actually emulating an SSP2-4.5 trajectory. Finally, if one wishes to select a single

tolerance to use for emulation of both SSP2-4.5 and SSP3-7.0 (and likely for similar, novel, intermediate scenarios), the larger $Z_{cutoff}$ should be used. For example, if one wished for a single $Z_{cutoff}$ value for CanESM5, $Z_{cutoff} = 0.13$ would provide generated ensemble sizes of 25 each for both SSP2-4.5 and SSP3-7.0 (with a draw-averaged max $Er$ of 5.1% rather than 4.3%).

By comparing the generated ensemble size from Table 5 with the CMIP6 archive sizes outlined in Table 1, we see that for

most ESMs, STITCHES can generate a stitched ensemble of the same size as the target ensemble. The cases with large archive sizes (CanESM5, MIROC-ES2L, MIROC6, and MPI-ESM1-LR) however, make it clear that the size of the stitched ensemble is not necessarily a direct function of the availability of runs in the archive or the size of the target ensemble, but depends on





the proximity in $(T, dT * X)$ space of the target windows to the archive windows. A look at the panels in Figure 1 gives a good representation of the challenges, as SSP370 appears to lie comfortably within the envelope of the SSP585 runs, whereas

SSP126 and SSP245 appear more isolated. We will discuss the implications of this in Section 4. The $Z_{cutoff}$ values in Table 5 are also not the final limits on where good matches may be generated. Specifically, because we start the matching neighborhood for each target point by finding the nearest neighbor in the archive first and then adding the tolerance to that distance (step 5 of the algorithm), there is a heterogeneity of matching neighborhood size for each target point even within a single trajectory. A different choice could uncover further results, however the choice to begin with nearest neighbors was made for convenience:

at $Z = 0$, the stitched trajectory returned is simply made up of the nearest neighbor points in the archive. Finally, there is utility in the stochastic draws used in this exercise as well. Multiple draws of generated ensembles fed through impacts models may lead to new insights, despite the fact that appending multiple draws together into a single "super"-generated ensemble is not advised due to envelope collapse.





**Table 5.** For each ESM and the two scenarios targeted by the emulation, we show the size of the archive, the number of trajectories used as target, and the number of stitched trajectories obtained from them, for the value of $Z_{cutoff}$ which keeps the metric $Er$, when averaged across 50 draws, at the maximum value indicated. We refer to Section 3.1.3 for details.

| Model | Target Scenario | Archive size | Target size | Stitched size | $Z_{cutoff}$ | $Er^*$ |
|---|---|---|---|---|---|---|
| ACCESS-CM2 | SSP2-4.5 | 16 | 3 | 5 | 0.175 | 5.4 % |
| BCC-CSM2-MR | SSP2-4.5 | 4 | 1 | 1 | 0.04 | 0.49% |
| CanESM5 | SSP2-4.5 | 100 | 25 | 25 | 0.105 | 4.3% |
| CAS-ESM2-0 | SSP2-4.5 | 8 | 2 | 3 | 0.265 | 4.6% |
| CESM2-WACCM | SSP2-4.5 | 10 | 3 | 3 | 0.115 | 5.4% |
| CMCC-CM2-SR5 | SSP2-4.5 | 4 | 1 | 1 | 0.04 | 1.7% |
| CMCC-ESM2 | SSP2-4.5 | 4 | 1 | 1 | 0.04 | 0.78% |
| FGOALS-g3 | SSP2-4.5 | 16 | 4 | 5 | 0.105 | 3.5% |
| FIO-ESM-2-0 | SSP2-4.5 | 9 | 3 | 3 | 0.09 | 3.4% |
| HadGEM3-GC31-LL | SSP2-4.5 | 9 | 4 | 1 | 0.04 | 2.2% |
| MCM-UA-1-0 | SSP2-4.5 | 4 | 1 | 1 | 0.04 | 1.1% |
| MIROC-ES2L | SSP2-4.5 | 60 | 30 | 66 | 0.215 | 5.9% |
| MIROC6 | SSP2-4.5 | 123 | 20 | 19 | 0.21 | 6.7% |
| MPI-ESM1-2-HR | SSP2-4.5 | 16 | 2 | 3 | 0.075 | 3.2% |
| MPI-ESM1-2-LR | SSP2-4.5 | 40 | 10 | 12 | 0.135 | 5.5% |
| NorESM2-LM | SSP2-4.5 | 7 | 3 | 4 | 0.125 | 4.6% |
| NorESM2-MM | SSP2-4.5 | 5 | 2 | 4 | 0.15 | 5.7% |
| UKESM1-0-LL | SSP2-4.5 | 37 | 6 | 13 | 0.18 | 4.7% |
| ACCESS-CM2 | SSP3-7.0 | 16 | 5 | 5 | 0.13 | 4.5 % |
| BCC-CSM2-MR | SSP3-7.0 | 4 | 1 | 1 | 0.04 | 0.36% |
| CanESM5 | SSP3-7.0 | 100 | 25 | 25 | 0.13 | 5.0% |
| CAS-ESM2-0 | SSP3-7.0 | 8 | 2 | 3 | 0.26 | 5.4% |
| CESM2-WACCM | SSP3-7.0 | 10 | 3 | 1 | 0.04 | 0.67% |
| CMCC-CM2-SR5 | SSP3-7.0 | 4 | 1 | 1 | 0.04 | 0.31% |
| CMCC-ESM2 | SSP3-7.0 | 4 | 1 | 1 | 0.04 | 0.75% |
| FGOALS-g3 | SSP3-7.0 | 16 | 4 | 5 | 0.12 | 0.54% |
| MCM-UA-1-0 | SSP3-7.0 | 4 | 1 | 1 | 0.04 | 0.3% |
| MIROC-ES2L | SSP3-7.0 | 60 | 10 | 28 | 0.255 | 3.9% |
| MIROC6 | SSP3-7.0 | 123 | 3 | 51 | 0.22 | 6.0% |
| MPI-ESM1-2-HR | SSP3-7.0 | 16 | 10 | 9 | 0.07 | 4.4% |
| MPI-ESM1-2-LR | SSP3-7.0 | 40 | 10 | 12 | 0.14 | 5.2% |
| NorESM2-LM | SSP3-7.0 | 7 | 3 | 1 | 0.04 | 2.1% |
| NorESM2-MM | SSP3-7.0 | 5 | 1 | 1 | 0.04 | 0.33% |
| UKESM1-0-LL | SSP3-7.0 | 37 | 13 | 13 | 0.16 | 5.4% |



## 4   Discussion and conclusions

We have proposed an algorithm, STITCHES, that exploits available simulations of future scenarios to deliver fully consistent and complete ESM output according to a new trajectory of global temperature over the 21$^{st}$ century. STITCHES works by stitching together decade-long windows (we use 9 years to be precise, but the length of the window is a tunable parameter) of existing 21$^{st}$ century simulations when GSAT matches in absolute value, $T$, and derivative, $dT$, the GSAT of a target scenario, similarly split into decade-long windows.

If the target scenario lies within a neighborhood populated by an archive of existing ESM simulations, in the space spanned by the two dimensions of the global warming level and the rate at which GSAT is changing, the algorithm can emulate it. The same algorithm can also be used to enrich the size of existing initial condition ensembles. In this case the target scenario may be part of the available simulations providing the building blocks. We have demonstrated the algorithm performance using the PANGEO CMIP6/ScenarioMIP archive of the four Tier 1 experiments, SSP1-2.6, SSP2-4.5, SSP3-7.0, SSP5-8.5, targeting the

emulation of the two intermediate scenarios.

Our validation tests have shown that the new trajectories do not reveal in the great majority of cases evident spurious behavior, even when the matching criteria are set to a generic value that is not specifically tailored to the internal variability of the ESM to be emulated. We have shown that jumps or discontinuities are seldom created at global scale, when considering surface temperature. Since surface temperature is the smoothest quantity among the variables commonly used to drive impact

models, we expect that any other variable at global or regional scale, and for yearly frequencies or higher, would be even more well-behaved at the seams, since the higher degree of internal variability would easily overwhelm discontinuities introduced by STITCHES. We have confirmed this for a few examples of gridded temperature and precipitation at the monthly frequency. We have also shown that for ENSO, arguably a very salient mode of variability for many natural and human systems, a 9-year window does not introduce odd frequency artifacts in the SOI time series. This should reassure modelers of impacts sensitive to

ENSO teleconnections. Synthetic "large ensembles" created to enrich initial condition experiments show an ensemble behavior within a small neighborhood of the truth (in most cases much narrower than $\pm 25 - 30\%$ of the target ensemble variability) in terms of ensemble mean and ensemble variance.

Our exploration of the performance of the algorithm as a function of the available archive size suggests that five 21st century trajectories ensure an acceptable performance (according to our metrics) and even smaller archive size often - if

not always – deliver acceptable stitched new trajectories. Thus, for modeling centers choosing to invest resources in future scenario simulations, running a well-chosen small set of trajectories that span what the community considers the plausible range of GSAT absolute change and rates of change, or radiative forcing, could suffice, and the center will be better served by focusing on running a few initial condition ensemble members for each trajectory, rather than investing in multiple similarly shaped scenarios. This also entails savings for the community that provides the direct forcing inputs to ESMs, translating IAM

output into spatially and time resolved forcing fields for future simulations. Resources in post-processing of model output, extending to the need of downscaling and bias-correcting, will be saved as well, as the emulated scenarios can be built from those post-processed ones.



Of course, our proposal does come with caveats. If ENSO frequencies are right around the timescale that is preserved by 9-year windows, there exist slower modes of variability in the climate system whose single phases may instead align with such time span, and whose coherent behavior would be broken by our window splitting and stitching together. Thus, any investigation of impacts that are known to be sensitive to low-frequency variability at decadal time scales needs to proceed with caution, try lengthening the window $X$, or not use STITCHES output at all. Similarly, any impact that depends on quantities whose integral is important, rather than their instantaneous value, cannot use the output from STITCHES if such integral frequently, or by definition, extends over the window size. Pre-eminently, sea level rise derived from ocean heat content, which is a path dependent quantity, cannot use the ocean heat content that comes with a STITCHES scenario, which would not be coherent with the scenario path. Similarly, mega-droughts lasting over a decade cannot be coherently represented in a scenario emulated by STITCHES.

There are more subtle aspects of stitched scenarios that may pose questions of fidelity and representativeness. They have to do with regional and or short-lived forcers like land-use and aerosols that usually vary across scenarios. STITCHES would not represent closely these forcers if the scenario to be emulated contained different regional patterns of them, compared to the scenarios used to generate the pieces to be stitched together. Thus, if those regional, short-lived forcers create climate signals that significantly alter the nature of the scenario they appear in, STITCHES would not replicate those signals. This is, however, not different from what happened in any analysis using time-sampling (King et al., 2018), or simple pattern scaling. Thus, here we work under the assumption that – amidst the uncertainties of different ESM responses and impact modeling affecting regional climate and impact outcomes – the signals introduced by regional and/or short-lived forcers would not be consequential to the results. We do encourage deeper exploration of these questions.

Last, some technical aspects of our algorithm will benefit from further analysis/considerations: possibly some applications may be able to relax the tolerance parameter, and thus set the conditions for easier matching and more numerous stitched realizations. This might be true of applications that won't be too sensitive to interannual differences. On the contrary, tightening the tolerance to match specific ESMs' internal variability will be beneficial in eliminating spurious behavior that we have documented in some cases, especially when the archive of available runs is poor.

We would have liked to make more than just a rule-of-thumb recommendation for the number of ensemble members that modeling centers should run, and link that formally to the number of expected trajectories created by STITCHES. That said, the last phases of CMIP have shown that, ultimately, modeling centers will commit what they can to running future scenarios. Our proposal shifts those energies and resources towards initial condition ensembles rather than different scenarios. One additional possibility that we haven't explored is utilizing idealized experiments like 1% $CO_2$ among the building blocks, consistently with our discussion of the secondary relevance (until proved wrong) of forcing agents regionally differentiated.

A type of scenario that is becoming more and more prominent in the policy discourse is the overshoot, i.e., a scenario that presents a peak and decline in global average temperature. STITCHES cannot emulate an overshoot scenario if none is available in the archive, since there would be no available windows to stitch together with GSAT having a negative derivative. If a range of overshoots are sought, there is the need to run some cases with different steepness and length in order to provide building blocks of decreasing temperature at different rates.



Despite the warranted caveats, we believe that our proposal has desirable outcomes for the research communities occupied with climate, scenario and impact modeling. Impact and IAM modelers that want to assess impacts for scenarios other than
those that have been generated by ESMs, including endogenously generate forcing pathways within IAMs, could rely on STITCHES to fill the gaps, acquiring the same type of output, in all its complexity and refinement, that an ESM would provide. An 'on-line' application of STITCHES within an IAM simulation could allow modeling climate impacts within the evolving system that the IAM is modeling, and therefore represent fully consistent feedback loops between climate change drivers (emissions) and climate change impacts. The wider impact research community could choose from a larger set of trajectories,
and possibly, a larger set of initial condition ensembles than ESM ran. Climate modelers can limit the effort devoted to preparing inputs for, setting up, running and post-processing future scenarios to a few of those. We acknowledge here the richness of climate model output archives already at our disposal (CMIP5, CMIP6, SMILES) which right now provide a wide variety of building blocks. The next phases of CMIP could be still a rich source of building blocks, despite being produced according to a much simplified design. The challenge would lie in choosing the best set of runs to optimally populate the $(T, dT)$ space to
maximize the number and shape of attainable new trajectories from the existing ones.

*Code and data availability.* The STITCHES software is available via GitHub (https://github.com/JGCRI/stitches/releases/tag/v0.9.0) and is frozen on zenodo (https://doi.org/10.5281/zenodo.6463264). Note that at the time of archiving, GitHub-zenodo integration was not functioning and so the pre-release STITCHES files were uploaded to zenodo directly. The code using the STITCHES software to generate data and the code analyzing data for this paper is available at a GitHub metarepository (https://github.com/JGCRI/Tebaldi_etal_2022_ESD)
and is frozen on zenodo(https://doi.org/10.5281/zenodo.6463270). All our ESM data is from the CMIP6 archive available through PANGEO (http://gallery.pangeo.io/repos/pangeo-gallery/cmip6/). The data generated using the STITCHES package and analyzed in this paper is archived (https://doi.org/10.5281/zenodo.6461693).





## Appendix A: Additional GSAT time series for intermediate scenarios

**Figure A1.** Examples of target (black lines) and stitched (colored) GSAT time series for ESMs in the PANGEO archive that ran at least one trajectory along the Tier 1 experiments of ScenarioMIP (SSP1-2.6; SSP2-4.5; SSP3-7.0; SSP5-8.5). We use the two bracketing scenarios to stitched together trajectories that follow the two intermediate scenarios, and compare true and stitched. As CMIP6 experiments were designed to branch multiple scenarios from the same historical simulation, when the model provides only few ensemble members in the archive chances are that some (at times, many) of the historical windows find the closest matches to be "themselves", only labelled SSP1-2.6 or SSP5-8.5. That explains the perfect match in the historical periods between traget and stitched trajectories. We do not worry about these cases, as the focus of our exercise is the creation of future scenarios.





**Figure A2.** Examples of target (black lines) and stitched (colored) GSAT time series for ESMs in the PANGEO archive that ran at least one trajectory along the Tier 1 experiments of ScenarioMIP (SSP1-2.6; SSP2-4.5; SSP3-7.0; SSP5-8.5). We use the two bracketing scenarios to stitched together trajectories that follow the two intermediate scenarios, and compare true and stitched. As CMIP6 experiments were designed to branch multiple scenarios from the same historical simulation, when the model provides only few ensemble members in the archive chances are that some (at times, many) of the historical windows find the closest matches to be "themselves", only labelled SSP1-2.6 or SSP5-8.5. That explains the perfect match in the historical periods between traget and stitched trajectories. We do not worry about these cases, as the focus of our exercise is the creation of future scenarios.



**Figure A3.** Examples of target (black lines) and stitched (colored) GSAT time series for ESMs in the PANGEO archive that ran at least one trajectory along the Tier 1 experiments of ScenarioMIP (SSP1-2.6; SSP2-4.5; SSP3-7.0; SSP5-8.5). We use the two bracketing scenarios to stitched together trajectories that follow the two intermediate scenarios, and compare true and stitched. As CMIP6 experiments were designed to branch multiple scenarios from the same historical simulation, when the model provides only few ensemble members in the archive chances are that some (at times, many) of the historical windows find the closest matches to be "themselves", only labelled SSP1-2.6 or SSP5-8.5. That explains the perfect match in the historical periods between traget and stitched trajectories. We do not worry about these cases, as the focus of our exercise is the creation of future scenarios.







**Figure A4.** Examples of target (black lines) and stitched (colored) GSAT time series for ESMs in the PANGEO archive that ran at least one trajectory along the Tier 1 experiments of ScenarioMIP (SSP1-2.6; SSP2-4.5; SSP3-7.0; SSP5-8.5). We use the two bracketing scenarios to stitched together trajectories that follow the two intermediate scenarios, and compare true and stitched. As CMIP6 experiments were designed to branch multiple scenarios from the same historical simulation, when the model provides only few ensemble members in the archive chances are that some (at times, many) of the historical windows find the closest matches to be "themselves", only labelled SSP1-2.6 or SSP5-8.5. That explains the perfect match in the historical periods between traget and stitched trajectories. We do not worry about these cases, as the focus of our exercise is the creation of future scenarios.



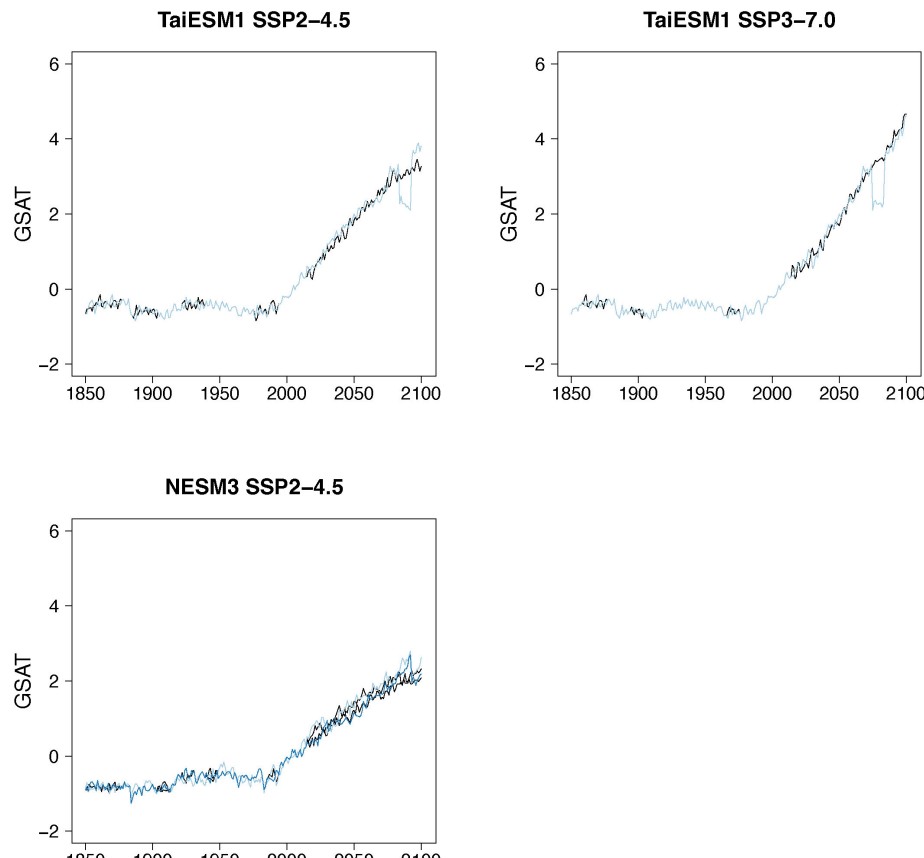

**Figure A5.** Examples of target (black lines) and stitched (colored) GSAT time series for ESMs in the PANGEO archive that ran at least one trajectory along the Tier 1 experiments of ScenarioMIP (SSP1-2.6; SSP2-4.5; SSP3-7.0; SSP5-8.5). We use the two bracketing scenarios to stitched together trajectories that follow the two intermediate scenarios, and compare true and stitched. As CMIP6 experiments were designed to branch multiple scenarios from the same historical simulation, when the model provides only few ensemble members in the archive chances are that some (at times, many) of the historical windows find the closest matches to be "themselves", only labelled SSP1-2.6 or SSP5-8.5. That explains the perfect match in the historical periods between traget and stitched trajectories. We do not worry about these cases, as the focus of our exercise is the creation of future scenarios.




## Appendix B: Additional trend and variability analysis of gridded data

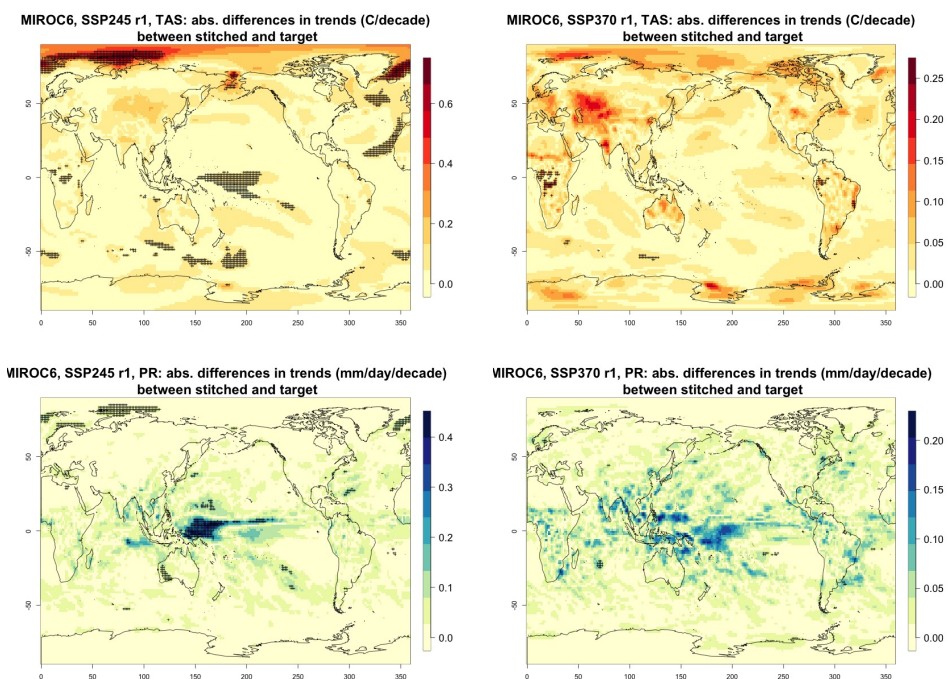

**Figure B1.** Absolute difference in decadal trends of temperature (TAS) and precipitation (PR) between stitched and target realizations. The value of the difference is expressed by the color scale and we marked as significant by black crosses those locations where the 95% confidence intervals of the trends computed from target and stitched time series do not overlap, indicating statistically significant differences. Emulation of MIROC6, monthly time series over 2015-2100, for SSP2-4.5 and SSP3-7.0. First realization.

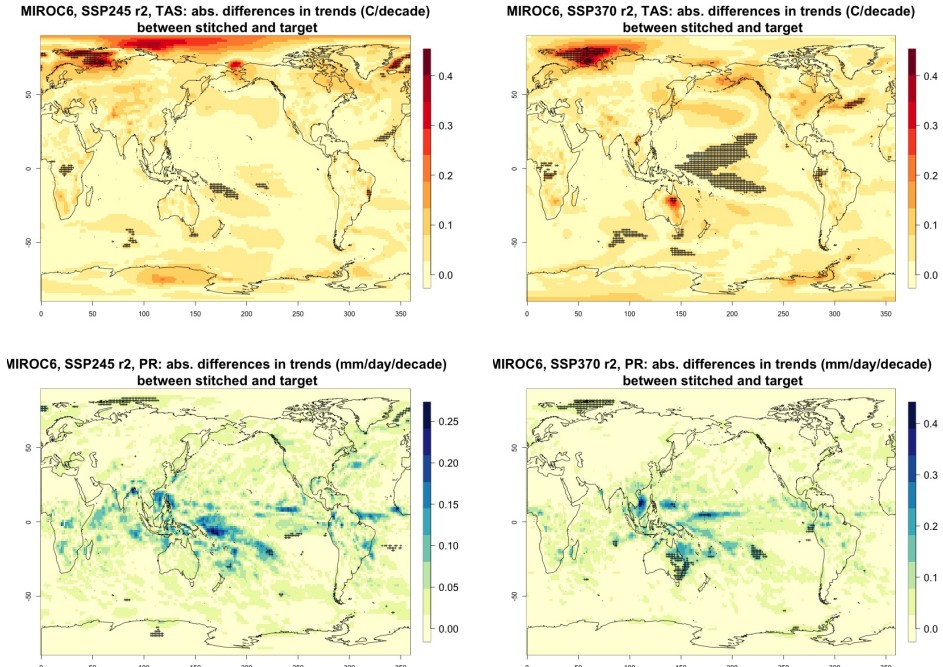

**Figure B2.** Absolute difference in decadal trends of temperature (TAS) and precipitation (PR) between stitched and target realizations. The value of the difference is expressed by the color scale and we marked as significant by black crosses those locations where the 95% confidence intervals of the trends computed from target and stitched time series do not overlap, indicating statistically significant differences. Emulation of MIROC6, monthly time series over 2015-2100, for SSP2-4.5 and SSP3-7.0. Second realization.



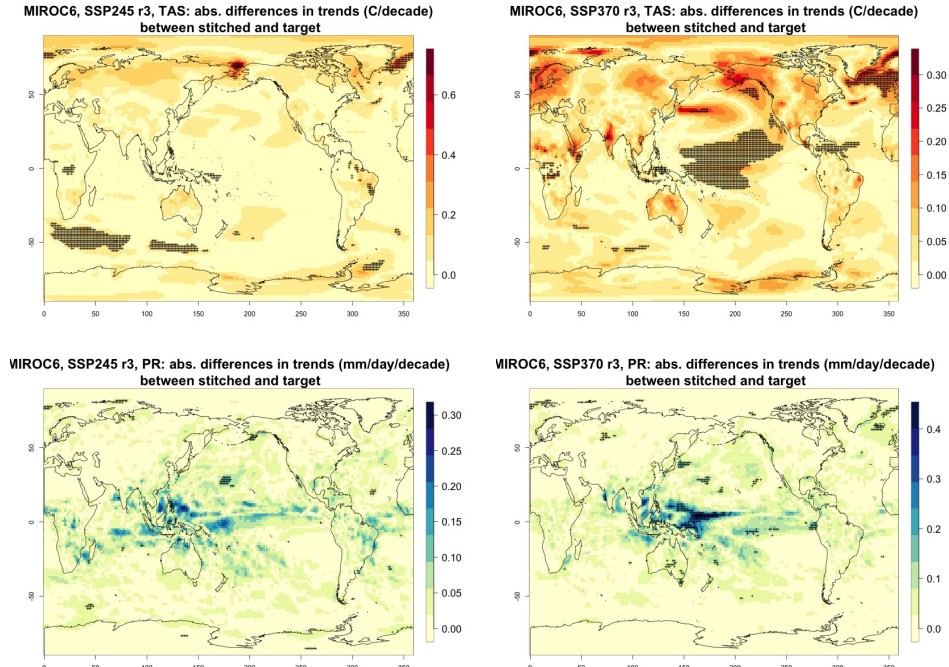

**Figure B3.** Absolute difference in decadal trends of temperature (TAS) and precipitation (PR) between stitched and target realizations. The value of the difference is expressed by the color scale and we marked as significant by black crosses those locations where the 95% confidence intervals of the trends computed from target and stitched time series do not overlap, indicating statistically significant differences. Emulation of MIROC6, monthly time series over 2015-2100, for SSP2-4.5 and SSP3-7.0. Third realization.



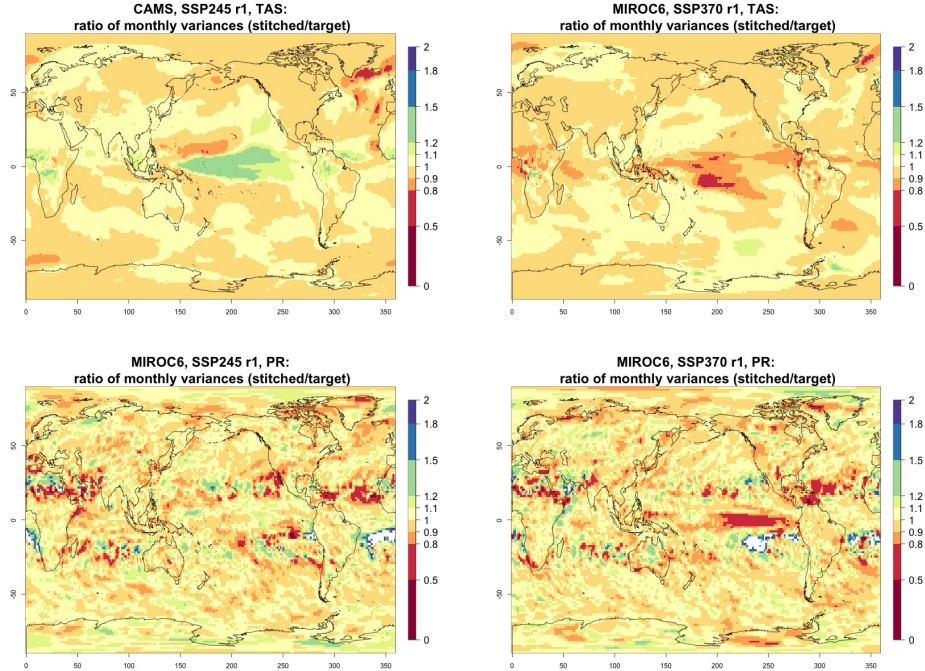

**Figure B4.** Ratio in monthly variability (standard deviation of residuals from trends) of temperature (TAS) and precipitation (PR) between stitched (at the numerator) and target (at the denominator) time series. The value of the ratio is expressed by the color scale which highlights the transitions at 0.8 and 1.2. Emulation of MIROC6, monthly time series over 2015-2100, for SSP2-4.5 and SSP3-7.0. First realization.



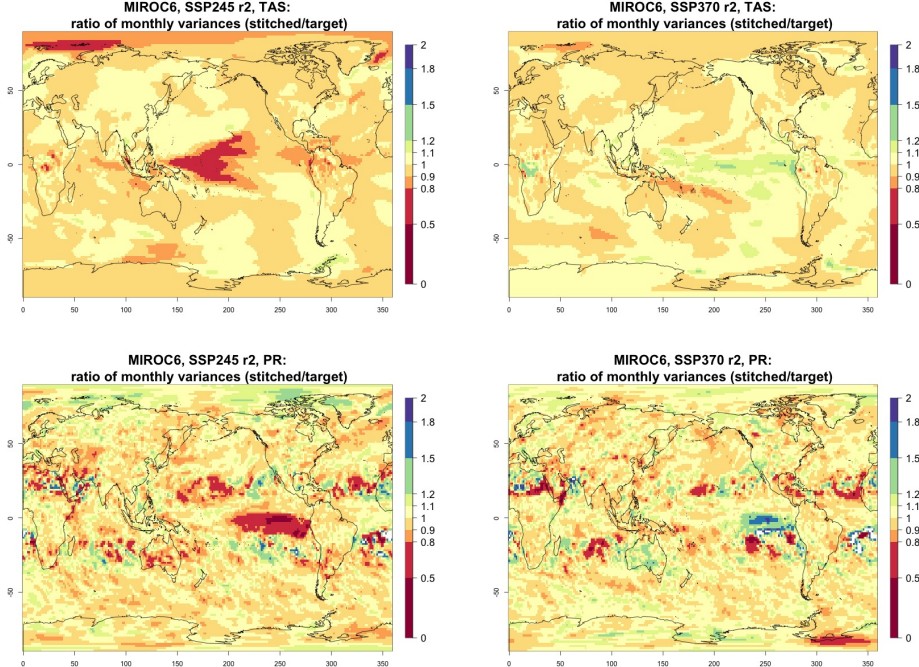

**Figure B5.** Ratio in monthly variability (standard deviation of residuals from trends) of temperature (TAS) and precipitation (PR) between stitched (at the numerator) and target (at the denominator) time series. The value of the ratio is expressed by the color scale which highlights the transitions at 0.8 and 1.2. Emulation of MIROC6, monthly time series over 2015-2100, for SSP2-4.5 and SSP3-7.0. Second realization.




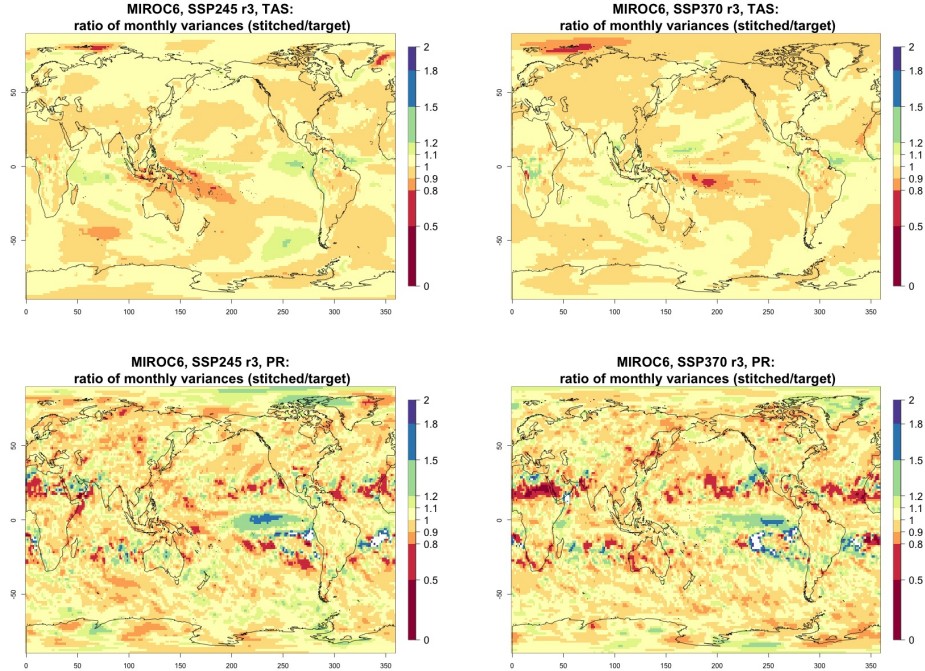

**Figure B6.** Ratio in monthly variability (standard deviation of residuals from trends) of temperature (TAS) and precipitation (PR) between stitched (at the numerator) and target (at the denominator) time series. The value of the ratio is expressed by the color scale which highlights the transitions at 0.8 and 1.2. Emulation of MIROC6, monthly time series over 2015-2100, for SSP2-4.5 and SSP3-7.0. Third realization.



## Appendix C: Additional SOI analysis for MIROC6


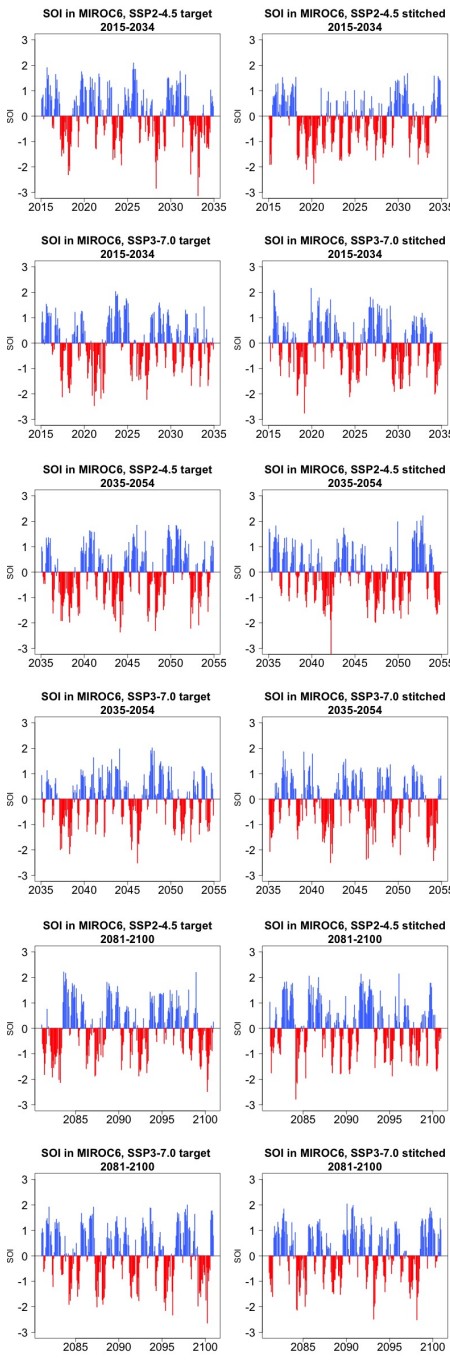

**Figure C1.** Examples of target (left) and stitched (right) SOI time series for three twenty-year windows along the length of the simulation: 2015-2034 in the top four panels; 2035-2054 in the middle four panels; 2081-2100 in the bottom four panels. Results from emulation of SSP2-4.5 and SSP3-7.0 for one of three ensemble members emulated under each scenario for MIROC6.


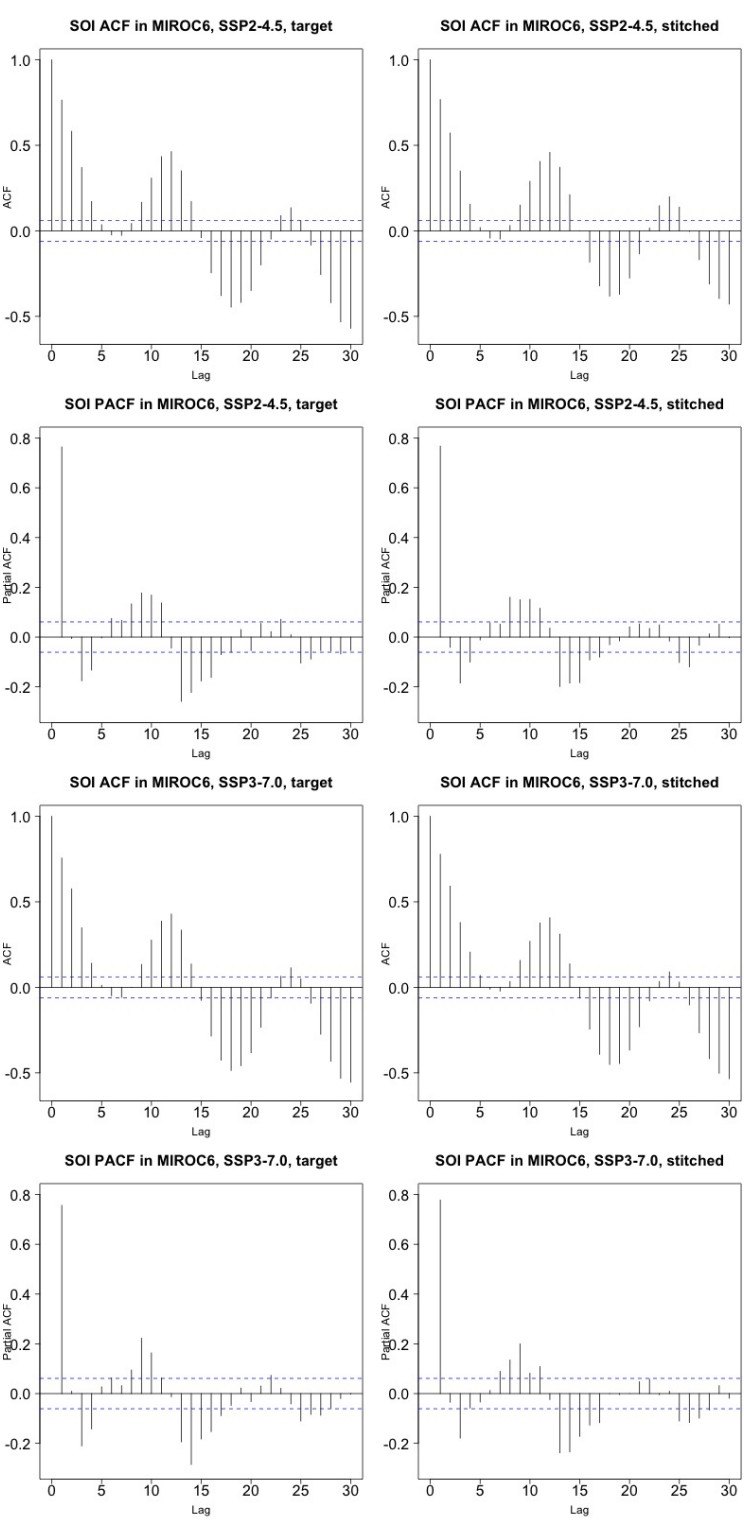

**Figure C2.** Auto-correlation (ACFs) and Partial auto-correlation functions (PACFs) for real and stitched SOI time series. Top two rows: SSP2-4.5 ACF for target and stitched series and respective PACFS. Bottom two rows: SSP3-7.0 ACF for target and stitchedseries and respective PACFs. Results from emulation of one of three ensemble members emulated under each scenario for MIROC6.



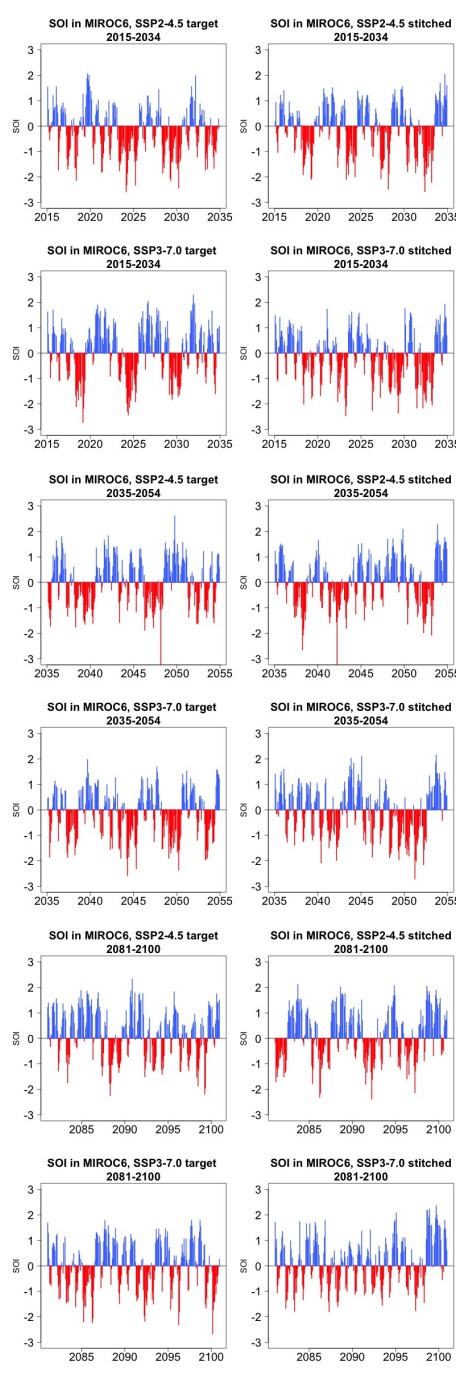

**Figure C3.** Examples of target (left) and stitched (right) SOI time series for three twenty-year windows along the length of the simulation: 2015-2034 in the top four panels; 2035-2054 in the middle four panels; 2081-2100 in the bottom four panels. Results from emulation of SSP2-4.5 and SSP3-7.0 for one of three ensemble members emulated under each scenario for MIROC6.



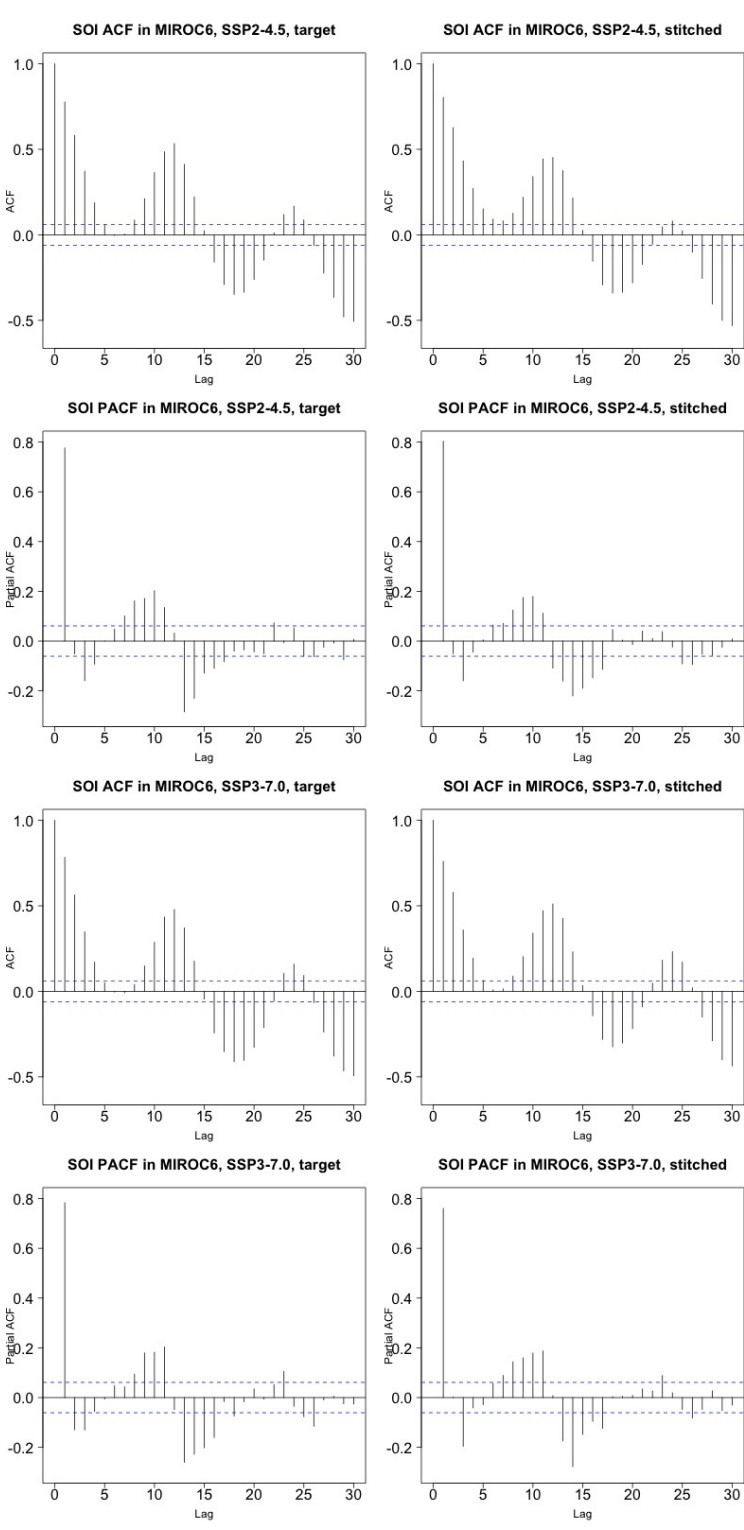

**Figure C4.** Auto-correlation (ACFs) and Partial auto-correlation functions (PACFs) for real and stitched SOI time series. Top two rows: SSP2-4.5 ACF for target and stitched series and respective PACFS. Bottom two rows: SSP3-7.0 ACF for target and stitchedseries and respective PACFs. Results from emulation of one of three ensemble members emulated under each scenario for MIROC6.





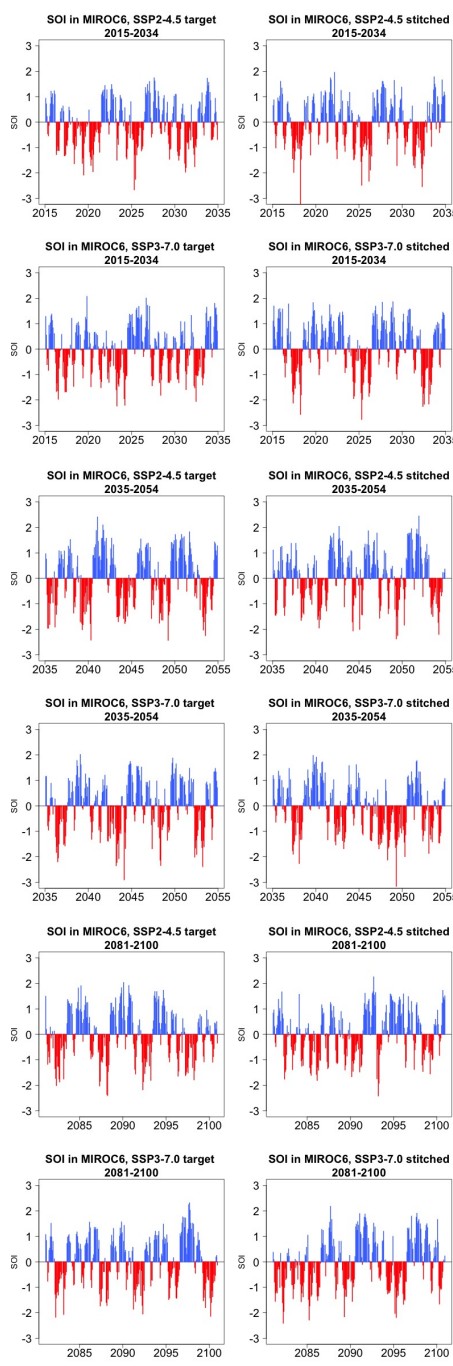

**Figure C5.** Examples of target (left) and stitched (right) SOI time series for three twenty-year windows along the length of the simulation: 2015-2034 in the top four panels; 2035-2054 in the middle four panels; 2081-2100 in the bottom four panels. Results from emulation of SSP2-4.5 and SSP3-7.0 for one of three ensemble members emulated under each scenario for MIROC6.

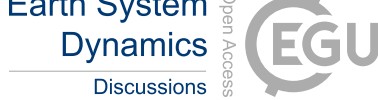

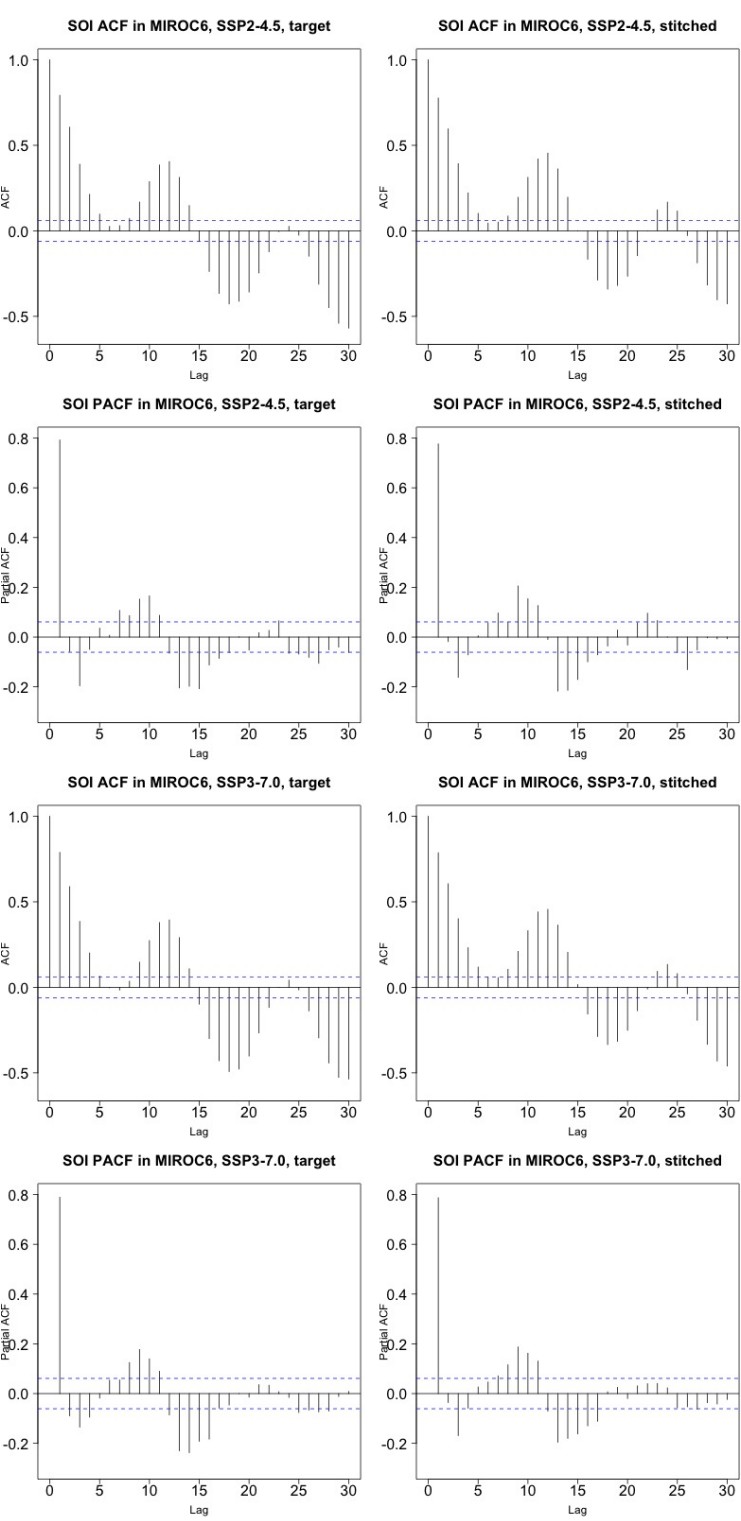

**Figure C6.** Auto-correlation (ACFs) and Partial auto-correlation functions (PACFs) for real and stitched SOI time series. Top two rows: SSP2-4.5 ACF for target and stitched series and respective PACFS. Bottom two rows: SSP3-7.0 ACF for target and stitchedseries and respective PACFs. Results from emulation of one of three ensemble members emulated under each scenario for MIROC6.





## Appendix D: GSAT time series for enriched ensembles

**Figure D1.** Examples of enriched ensembles of GSAT time series for ESMs in the PANGEO archive that have at least 5 trajectories available over the 21$^{st}$ century. As in the figures in Appendix A, warmer colors indicate a larger number of stitched trajectories in the figure, as the title also describes.





**Figure D2.** Examples of enriched ensembles of GSAT time series for ESMs in the PANGEO archive that have at least 5 trajectories available over the 21$^{st}$ century. As in the figures in Appendix A, warmer colors indicate a larger number of stitched trajectories in the figure, as the title also describes.





**Figure D3.** Examples of enriched ensembles of GSAT time series for ESMs in the PANGEO archive that have at least 5 trajectories available over the 21[st] century. As in the figures in Appendix A, warmer colors indicate a larger number of stitched trajectories in the figure, as the title also describes.





**Figure D4.** Examples of enriched ensembles of GSAT time series for ESMs in the PANGEO archive that have at least 5 trajectories available over the 21$^{st}$ century. As in the figures in Appendix A, warmer colors indicate a larger number of stitched trajectories in the figure, as the title also describes.



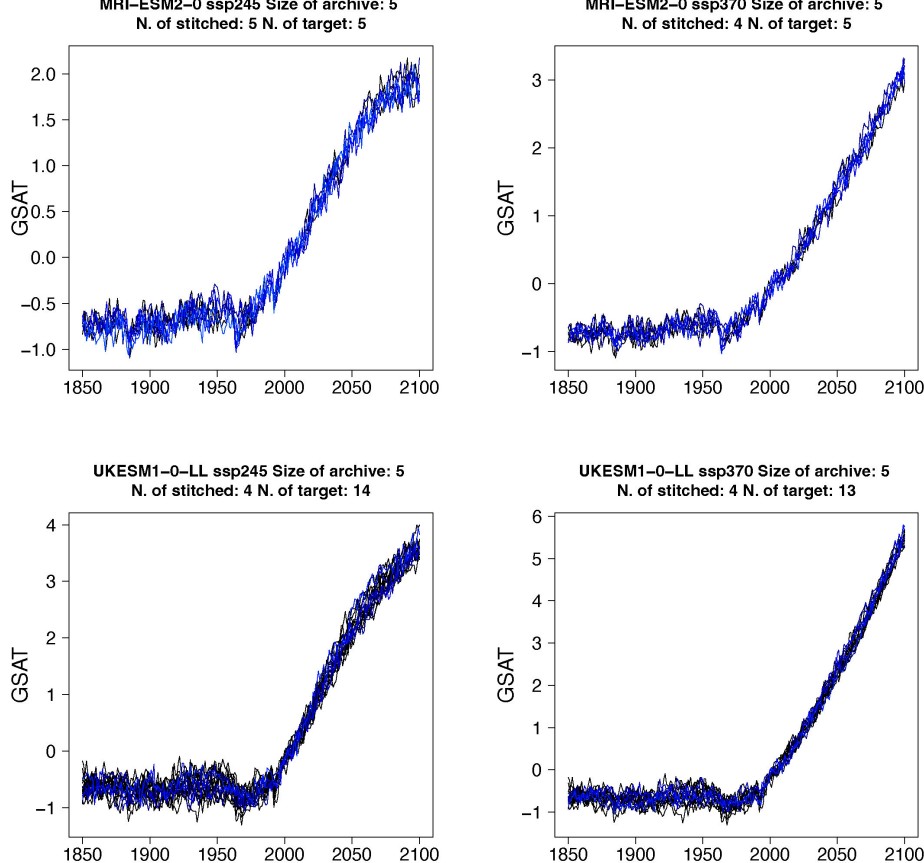

**Figure D5.** Examples of enriched ensembles of GSAT time series for ESMs in the PANGEO archive that have at least 5 trajectories available over the 21$^{st}$ century. As in the figures in Appendix A, warmer colors indicate a larger number of stitched trajectories in the figure, as the title also describes.

**Appendix E: Table of $E1$ and $E2$ metrics for enriched ensemble exercise performed using only bracketing scenarios SSP1-2.6 and SSP5-8.5.**



**Table E1.** The two components of the $Er$ metric, $E_1$ and $E_2$, computed for several experiments across ESMs, scenarios and number of available archive trajectories from which to create the stitched ensembles. Numbers in columns 4 through 9 represent fractions of the target ensemble variance (see formula 1).

| Model | Scenario | Archive Size | Target Members | Stitched Members | $E_1$ 2010 | $E_1$ 2050 | $E_1$ 2090 | $E_2$ 2010 | $E_2$ 2050 | $E_2$ 2090 |
|---|---|---|---|---|---|---|---|---|---|---|
| ACCESS-CM2 | ssp245 | 5 | 5 | 4 | 0.07 | 0.12 | 0.38 | 1.05 | 0.50 | 2.01 |
| ACCESS-ESM1-5 | ssp245 | 5 | 10 | 1 | 0.68 | 0.18 | 0.02 | 1.20 | 0.67 | 0.75 |
| CanESM5 | ssp245 | 5 | 25 | 2 | 0.10 | 0.44 | 1.24 | 1.20 | 1.09 | 6.94 |
| MIROC-ES2L | ssp245 | 5 | 30 | 2 | 0.00 | 2.57 | 0.61 | 0.96 | 1.43 | 0.70 |
| MPI-ESM1-2-LR | ssp245 | 5 | 10 | 2 | 0.47 | 0.06 | 1.22 | 1.31 | 1.50 | 0.77 |
| MRI-ESM2-0 | ssp245 | 5 | 5 | 1 | 0.50 | 0.09 | 6.34 | 0.75 | 1.76 | 2.49 |
| UKESM1-0-LL | ssp245 | 5 | 14 | 1 | 0.04 | 0.00 | 6.57 | 0.47 | 0.56 | 1.69 |
| ACCESS-ESM1-5 | ssp245 | 10 | 10 | 3 | 0.06 | 0.06 | 0.55 | 0.55 | 1.57 | 1.43 |
| CanESM5 | ssp245 | 10 | 25 | 5 | 0.00 | 0.00 | 3.19 | 1.41 | 0.81 | 7.67 |
| MIROC-ES2L | ssp245 | 10 | 30 | 2 | 1.53 | 0.90 | 0.24 | 0.82 | 1.06 | 2.06 |
| MPI-ESM1-2-LR | ssp245 | 10 | 10 | 5 | 0.06 | 0.09 | 0.42 | 0.84 | 1.10 | 2.43 |
| CanESM5 | ssp245 | 15 | 25 | 3 | 0.48 | 0.01 | 0.16 | 1.35 | 1.03 | 3.66 |
| CanESM5 | ssp245 | 20 | 25 | 6 | 0.01 | 0.18 | 5.27 | 1.07 | 1.09 | 5.22 |
| CanESM5 | ssp245 | 25 | 25 | 7 | 0.01 | 0.05 | 4.30 | 0.84 | 1.00 | 4.34 |
| ACCESS-CM2 | ssp370 | 5 | 5 | 3 | 0.07 | 0.06 | 0.06 | 1.90 | 2.61 | 1.34 |
| ACCESS-ESM1-5 | ssp370 | 5 | 30 | 4 | 0.07 | 0.52 | 0.11 | 0.51 | 1.23 | 1.13 |
| CanESM5 | ssp370 | 5 | 25 | 2 | 0.00 | 1.85 | 0.22 | 1.12 | 0.28 | 2.48 |
| MIROC-ES2L | ssp370 | 5 | 10 | 3 | 1.41 | 0.27 | 0.14 | 0.91 | 1.63 | 1.57 |
| MPI-ESM1-2-LR | ssp370 | 5 | 10 | 4 | 0.40 | 0.07 | 0.02 | 0.81 | 1.73 | 2.39 |
| MRI-ESM2-0 | ssp370 | 5 | 5 | 3 | 0.12 | 0.12 | 0.08 | 1.27 | 1.24 | 1.63 |
| UKESM1-0-LL | ssp370 | 5 | 13 | 2 | 0.21 | 0.30 | 0.09 | 0.70 | 0.84 | 1.78 |
| ACCESS-ESM1-5 | ssp370 | 10 | 30 | 5 | 0.07 | 0.00 | 0.01 | 0.96 | 1.24 | 2.06 |
| CanESM5 | ssp370 | 10 | 25 | 2 | 1.29 | 0.16 | 1.13 | 1.53 | 0.89 | 0.68 |
| MIROC-ES2L | ssp370 | 10 | 10 | 4 | 0.29 | 0.00 | 0.05 | 1.06 | 1.16 | 1.08 |
| MPI-ESM1-2-LR | ssp370 | 10 | 10 | 4 | 0.03 | 0.24 | 0.05 | 1.23 | 1.84 | 1.87 |
| CanESM5 | ssp370 | 15 | 25 | 8 | 0.00 | 0.15 | 0.81 | 0.67 | 2.07 | 1.94 |
| CanESM5 | ssp370 | 20 | 25 | 7 | 0.02 | 0.31 | 0.00 | 1.08 | 1.32 | 2.53 |
| CanESM5 | ssp370 | 25 | 25 | 6 | 0.00 | 0.00 | 1.04 | 1.04 | 1.31 | 0.84 |



*Author contributions.* CT conceived the general approach. AS and KD significantly refined it, trouble-shot it and implemented it in Python. All authors collaborated in testing the results. CT led this paper write-up with AS and KD contributing to it.

*Competing interests.* All authors declare no conflict of interest.

*Acknowledgements.* This work was conducted with the support of the U.S. Department of Energy, Office of Science, as part of research in MultiSector Dynamics, Earth and Environmental System Modeling Program. CT was also supported by the CASCADE project, funded by the U.S. Department of Energy, Office of Science, Office of Biological and Environmental Research, as part of the Regional and Global Modeling and Analysis Program. The Pacific Northwest National Laboratory is operated by Battelle for the US Department of Energy (contract no. DE-AC05-76RLO1830). Lawrence Berkeley National Laboratory is operated by the US Department of Energy (contract no. DE340AC02-05CH11231).





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
