# Peer review of "STITCHES: creating new scenarios of climate model output by stitching together pieces of existing simulations"

_Earth System Dynamics, 2022_

## Author Comment (AC1)

**Reviewer 1**

The STITCHES algorithm presents a unique time-sampling based approach that enables exploration of different, arbitrary climate scenarios. Its added benefit of not being limited to specific climate variables or spatial/temporal scales makes it a powerful tool in comparison to existing simple climate models/emulators. Overall, it is extremely relevant to the climate modelling and impact/integrated assessment societies and suitable for the Earth System Dynamics journal.

Thank you for your positive reception of this work and your careful and constructive review.

Some comments are as follows:

**High-level comments:**

1. The "outside the lower-end emission scenario bracket" application of STITCHES should be clarified, there is discussion surrounding overshoot however not for low-emission scenarios with near equilibrated climate by 2100.

We have modified our discussion throughout to include the lower end (or in general, extrapolation outside of the existing envelope) as a show stopper for STITCHES, together with the emulation of scenarios whose shape is not well represented at this time in the CMIP6 archive we are using as our sandbox. While emulating scenarios above the highest is definitely impossible for STITCHES, a scenario that is lower than the lowest available but still greater than or equal to historical levels of warming could be in theory emulated, but the meaning of such scenario could be argued as not exactly apparent.

We have added/reworded the last sentence of the abstract on these points: *Given that by definition STITCHES cannot emulate scenarios that result in GSAT trajectories outside of the envelope available in the archive, neither can it emulate trajectories with shapes different from existing ones (overshoots with negative derivative, for example) the size and characteristics of the available archives are the principal limitations of STITCHES deployment. Thus, we argue for the possibility of designing scenario experiments within, for example, the next phase of the Coupled Model Intercomparison Project according to new principles, relieved of the need to produce a number of similar trajectories that vary only in radiative forcing strength, but more strategically covering the space of temperature anomalies and rates of change.*

2. Some discussion on choice of tuning parameters (X and Z) for different temporal scales (annual vs monthly) should also be given. Since non-linear warming could

manifest more strongly at monthly timescales (due to e.g. snow-albedo feedbacks), this could limit the values of X or Z to be used (or otherwise the fineness of temporal resolution). Given that decadal oscillatory patterns such as El-Nino are aimed to be conserved, implications of having X>9 and the compromise this has on fidelity of representation for finer temporal resolutions should furthermore be explored (e.g. looking at performance on monthly timescales with different X values).

Perhaps we are misunderstanding the reviewer's point, here, but we think that monthly behavior would not be affected if not at the seams by a different choice of X and Z, given that once the sequence of pointers is created, the behavior of monthly variable is that of the original ESM output. The X and Z parameters apply to annual global temperature time series by construction, importantly because we want STITCHES to emulate scenarios on the basis of a trajectory of GSAT produced by simple models, which usually do not produce monthly output. Thus, X and Z, rather than reflecting on the behavior of monthly time series, are designed to ensure that what we are emulating is the forced component, and that we do not introduce severe discontinuities at the seams vis-a-vis the behavior of slower (multi-annual) modes of variability. Post-facto we do not encounter many cases where the behavior of monthly variables shows artifacts from the stitching, as documented in the validation section of our paper. We have added however a sentence to the section looking at the choice of Z that mentions the possibility of considering that for values of X very different from what we use, 9. WE invite the users to do that, as we provide the software where both values are tunable parameters.

3. Although discussion of application of STITCHES is given, readers would be curious for more discussion on future developments and improvements that could be made.

We think we can address the reviewer suggestion both by pointing at possible developments of our algorithm itself (for example, alternative choices of metrics for the nearest-neighbor space and distance) and importantly about what we see as promising developments in the field, with plans to join forces with other type of emulators, like MESMER-M and MESMER-X and a recently submitted emulator proposal, called PREMU.

In particular, within the last section of the paper we have added a sentence related to possible modification of the technical aspects:

Last, some technical aspects of our algorithm will benefit from further analysis/considerations: possibly some applications may be able to relax the tolerance parameter, and thus set the conditions for easier matching and more numerous stitched realizations. This might be true of applications that would not be too sensitive

to interannual differences. On the contrary, tightening the tolerance to match specific ESMs' internal variability will be beneficial in eliminating spurious behavior that we have documented in some cases, especially when the archive of available runs is poor. *More generally we could choose a difference distance measure in the $(T,X*dT)$ space, or a completely different space over which looking for nearest neighbors, but the necessity of conforming to what a simple model can produce on the basis of a new emission scenario needs to be kept as a consideration.*

We have also concluded the paper with an explicit call for using the novel emulators that are being developed of late in a complementary manner:

*The deployment of STITCHES, in concert with other emulators like MESMER-M and X~\cite{beuschetal2020,beuschetal2021,Quilcailleetal2022} and PREMU~\cite{Liuetal2022}, which are intended to produce new realizations of internal variability could then complement and enrich the effort of the ESM community.*

With a new citation for a paper in discussion at the moment proposing a new emulator for precipitation:

*Liu, G., Peng, S., Huntingford, C., and Xi, Y.: A new precipitation emulator (PREMU v1.0) for lower complexity models, Geosci. Model Dev. Discuss. [preprint], https://doi.org/10.5194/gmd-2022-144, in review, 2022.*

Below are more specific comments

**Specific comments:**

L4: the link between emulators and computational demand should be clarified

We have added a few words here in the abstract to this effect: Given the computational cost of running coupled Earth System Models (ESMs), *which are usually the domain of super computers and require on the order of weeks to complete a century-long simulation,* only a handful of different scenarios are usually explored by ESMs. An effective emulator, *able to run on standard computers in times of the order of minutes, rather than days,* could therefore be used to derive climate information under scenarios that were not run by ESMs.

L19: This may be confusing to readers: the use of GSAT to create the pointers from which all other climate variables at different spatial and temporal scales will be stitched together should be clarified (i.e. pointer is not climate variable specific).

Thank you for underlining this, it is really the crux, and we will make sure to clarify, also given the comments of Reviewer 3 which indicate the need of being more careful with the use of our terms and language, evidently confusing to some. In the abstract we have reworded now by specifying: A look-up table is therefore created of a sequence of existing windows/experiments that, when stitched together, create a GSAT trajectory "similar" to the target. *Importantly, we can then stitch together much more than GSAT from these windows, i.e., any output that the ESM has saved for these experiments/time windows, at any frequency and spatial scale available in its archive.*

L113: This suggestion is a bit strong given that emulators already mentioned (Link et al. 2019, Beusch et al. 2020,2021) circumvent the need for initial condition ensembles by providing stochastically generated imitations of the expected internal variability. Furthermore, scenario exploration to look at climate under equilibrated or overshoot state is still extremely important, and this should be clarified.

Absolutely agreed, and we have reworded this sentence altogether as: *If emulators, possibly used in a complementary fashion, become part of the overall strategy in providing climate information to the impact research community we argue that the next ScenarioMIP design may identify different priorities from the current one.*

L115-L135: Very well explained background to the rationale!

**Thank you.**

L146: what about scenarios lower than the lowest emission scenarios or overshoot scenarios?

We rephrased adding : We note here, however, that by construction our algorithm does not allow extrapolating to levels of warming above those of the highest scenario available in the archive, *or below the lowest.*

We initially did not worry about lower than the lowest, since the historical simulations would be the lower limit, and those are lower than the lowest, and available. Realistically though interesting scenarios lower than the lowest would be overshoots,

and for those our caveats about the lack of a rich-enough archive remains valid. We discuss this latter point later on, after introducing the (T, X*dT) space. We write: Note that when the goal is emulating non-existing scenarios, our targets need to be trajectories that reach warming levels within the ones available as building blocks in the archive, as our algorithm does not allow extrapolating. Similarly, STITCHES stops short of being able to emulate overshoot scenarios, given that the archive does not offer a large population of overshoot experiments that we can use as building blocks (i.e., the cooling behavior of GSAT in an overshoot experiment cannot be sampled from increasing, or flat, GSAT trajectories). These considerations could be useful to keep in mind when designing the next phase of ScenarioMIP.

L197-L205: Z is dependent on X which is also a tuning parameter, this may introduce additional caveats in choosing X so as to avoid "jumps" between the seams. Have sensitivity tests been performed on this? Some explanation on how to jointly pick the optimal combination of X and Z should be provided.

We have kept the two choices separate, as we worry about X in the context of adapting the smoothness of the archived/available GSAT series to the time series that wewould get from a simple model. We then have a session later in the paper that discusses our investigation of the sensitivity of the algorithm results to the choice of Z. Please see Section 3.1.3. Our goal is to publish the code where all these parameters can be tuned (to specific ESMs, and specific applications) rather than trying to come up with gold standards that we believe would be anyway sensitive to the two choices mentioned above. We do add a sentence however, in this section, inviting exploration of the choice of Z, depending on values of X.

L211: Is the ensemble size the sole thing considered when choosing which ESMs to display? Looking at ESMs of different genealogies would also be interesting especially for the (T, XdT) space (if not that is also O.K., just curious about why the above criteria).

We chose to develop our emulator on the basis of the  CMIP6/ScenarioMIP archive and are using all models that provided a subset of monthly and daily variables that we set out to emulate. Some of these models have very small ensemble sizes, some have large ensembles. It would be possible to look at past generations of models. We wonder if the reviewer is thinking about combining the archives across the same model's different versions. That, we think, would be problematic given how different successive versions of the same model can be. So we did not go there. Ideally, the same version of the model would have run both sets of scenarios and that would make

the archive richer, but we have not found that to be the case, with most ESMs having submitted a new version to the latest phase of CMIP.

Figure 1: it seems that for most models around -0.01degC the rate of historical warming is higher than that at 0-0.01 degC, is there a reason for this? It also raises the question of the generalizability of this approach for time windows with major volcanic events (e,g, Mt Pinatubo which has a distinct fingerprint in the GMT trajectory) and some elaboration on this may be required.

The reviewer has identified something that we did not notice, having focused our application to the scenario part (future) of the simulations and that indeed seems specific to volcanic eruptions. We have added a sentence in the conclusions pointing this out:

There are more subtle aspects of stitched scenarios that may pose questions of fidelity and representativeness. *We have not addressed the challenges that short but intense forcing episodes, like volcanic eruptions, may pose, since we have focused the application of STITCHES on future scenarios, which do not represent them. A careful look at Figure~\ref{fig:PANGEO_archive} can highlight a region of the space populated by grey dots (the historical part of the simulations) showing a peculiar pairing of absolute temperature anomalies and rate of change in the region around $T=-0.01$ compared to that around $T=0.01$. This would suggest a specific behavior of GSAT while recovering from volcanic eruptions that is not easily emulated by finding analogs in the historical period (away from volcanic episodes).*

L227-L230: Great that this is elaborated upon here! Providing this elaboration earlier could benefit and provide more structure to the text however.

We have added these points to the Introduction. Specifically, we added a sentence in the next to last paragraph:

We split the GSAT trajectory into regular windows, and we identify for each of them a "nearest neighbor" among the windows of GSAT trajectories available in the archive, from the finite number of experiments that were run and archived, *as long as the scenario that is target of our emulation is characterized by an intermediate level of GSAT warming, and similar rates of change to those present in the archive.*

And we pointed out explicitly overshoots and stabilized scenarios in the last paragraph:

*We also discuss the challenges that STITCHES encounters when targeting scenarios of shapes other than regularly increasing forcings, like stabilized scenarios and overshoots, therefore suggesting that a concerted effort in exploring scenarios of different shapes, rather than scenarios that only vary in the strength of the radiative forcing, could be made to facilitate the application of the emulator.*

Figure 2: It seems that all ESMs in this figure have a mismatch in the GSAT trajectories after 2050 for ssp 2-4.5 (and also BSS-CSM2-MR and CMCC-ESM2 in Figure 4), some elaboration on this may be needed e.g. transient vs equilibrated state. In general some consideration of how to stitch together cases where X*dT ~ 0 should be elaborated as nearest neighbors could have both a positive or a negative trend.

We have added a sentence, as suggested:

*Also from these figures one can assess that the behaviors that appear to deviate from the expected, are all at the tail end of the simulations, and only for those models that offer only one pair of scenarios in the archive to sample from, particularly for SSP2-4.5, which adds the extra challenge of a trajectory that stabilizes ($ dT \approx 0$) and needs to find matches among windows from scenarios that, at that level of warming, are by construction increasing in forcings. In general, stabilization scenarios together with overshoots pose a challenge to STITCHES given the content of the CMIP6 archive from which we construct our emulations.*

L306: It would be interesting to see month specific trends (e.g. the decadal trend for Jan and Jul). It seems here it is only the decadal trend of the whole monthly time series, if not this should be clarified as well.

We will provide these.

Figure 6: There seems to be systematic overestimation of monthly variance around central Africa (also for models in the appendix), are there reasons for this (e.g.

vegetation/land cover changes where SSP 5-8.5 imposes quite high deforestation which may lead to spurious variabilities)

First, we realized we had by mistake included a panel for this monthly TAS variability plot in the appendix for CAMS, rather than MIROC6. MIROC6 SSP2-4.5 does not show the same patches of overestimated variability over central Africa, while only the lower area is present for SSP3-7.0. It may be true that some effects of vegetation may be surfacing here but it would be fairly speculative of us to discuss this, also given the fact that, for CAMS, the pattern is the same for 4.5 and 7.0, that have different land use assumptions.

L321: The argument that internal variability explains the mismatch in the Arctic is not so convincing. It could for instance be due to the AMOC or otherwise due to a non-linear increase in summer time temperatures during ice-free arctic summers.

We identified internal variability as the explanation because the patch appears in two out of three ensemble members, but we are happy to add these other explanations as possibilities, as suggested:

Internal variability is likely responsible for an area in the Arctic appearing as inconsistent in two of the three realizations, *but effects of ice-free summer intensified warming, or behavior of the AMOC could contribute to this limited area of disagreement.*

L346: Figure 7, it may be difficult to visually gauge similarity in magnitude and oscillatory behaviour. Although this is made more obvious in Figure 8, it may be a good idea to apply a power spectral decomposition instead and show their results for a clearer overview. Very good idea to look at SOI within the analysis otherwise!

We will include spectra.

L400: Does the Z_cutoff value generalize to all values of X? The calculation of Z_cutoff is already a very useful exercise so this is a minor detail, just curious.

We haven't gone there but added a sentence pointing at possible exploration of this issue, enabled by our software, in Section 3.1.3 (about the sensitivity to choices of Z of the number of ensemble members obtainable).

L438: The term envelope collapse should be clarified and how it related to the Z value as well (i.e. how best to know at which Z envelope collapse has been approached?

We have rephrased the sentence where the term appears in the Methods section (step 6) to clarify its meaning and also added clarification to Section 3.1.3.

Table 5: Is there a relationship (e.g. linear) between between $E_r$ and $Z\_cutoff$, or are they stable and then jump to above 10% after a certain cutoff?

For a particular ESM, $E_r$ and $Z\_cutoff$ tend to increase together. However, due to the discrete nature of our matching set up between a finite number of target and archive windows, there are clear stable values followed by step increases in this relationship. Many values of Z can result in the same set of archive windows matching to a target window, until Z crosses some threshold and another archive window gets added to the set of matches. At a certain point, Z increases enough that the next added archive window is too different from the target window to be a 'good' match (and $E_r$ has a step increase to reflect that). This, combined with the fact that the specific generated ensemble members for a given Z value are stochastic, is why we select $Z\_cutoff$ via this post-hoc set of experiments rather than directly within the algorithm itself.

Table E1: The $E_1$ and $E_2$ values for CanESM5 tend to be higher for 20 archive members and then drop lower at 25 archive members. More so for SSP 3-7.0 the $E_1$ values are 0 at 25 archive members for both 2010 and 2050. Is there a reason for this?

We regard this table as showing fairly noisy results…we are submitting STITCHES to a tall order in having to emulate two scenarios multiple times on the basis of two bracketing, very different scenarios. The way we set up the exercise is by randomizing the members of the full archive (of CanESM5 in this case) included in the smaller ensembles (e.g., we choose 5, 10, 15, 20 members randomly from the 25 available). The algorithm also randomizes the choice of nearest neighbors. So, the patterns of these metrics are not easily interpretable. We would expect that if we repeated this exercise many times the average outcome would lend itself to a better interpretation, but this exercise is mostly about showing the strain imposed on the algorithm when supplying such extreme brackets.

We have added a sentence that highlights the noisy nature of these results when pointing at the table:

We have performed the same exercise by limiting the archive to the two bracketing scenarios, SSP1-2.6 and SSP5-8.5, and trying to construct ensembles for SSP2-4.5 and SSP3-7.0. In this case STITCHES is significantly challenged, and its performance as measured by the $E_r$ metric significantly diminished *and, when comparing what happens for the same model and increasing numbers of archive members, unpredictable, due to the fact that the algorithm randomizes both the identity of the archive members and the choice of the nearest neighbors to construct the emulated output.*

Conclusion and Discussion: the recommendation for looking at less scenarios and focusing on more initial condition ensembles may be quite strong: perhaps there should be elaboration on which scenarios are more useful to explore (i.e. ones where interpolation becomes difficult such as overshoot or equilibrated climate). The applicability of STITCHES across different temporal scales should also be clarified (i.e. limitations when applying it to annual vs monthly vs subdaily timescales).

We have modified our discussion of the implications for CMIP/ScenarioMIP by simply pointing at the need of populating the space of (T, dT) more effectively. We are also calling for the use of other type of emulators jointly with STITCHES.:

*The next phases of CMIP could complement what is available now by deliberately exploring types of scenarios that are not well represented in the current archives, like stabilized trajectories and overshoots. The challenge would lie in choosing the best set of runs to optimally populate the $(T,X*dT)$ space to maximize the number and shape of attainable new trajectories from the existing ones. The deployment of STITCHES, in concert with other emulators like MESMER-M and X~\cite{beuschetal2020,beuschetal2021,Quilcailleetal2022} and PREMU~\cite{Liuetal2022}, which are intended to produce new realizations of internal variability could then complement and enrich the effort of the ESM community.*

**Editorial comments:**

L35: support the climate information needs of the impact research community

L44: bias-correcting them. Alternatively just bias-correction could also work

L120: perhaps "scenario-independence" would be a term more consistent with the terms already introduced

L147: "the STITCHES algorithm"

Figure 1: Lovely plots, very informative! Font size needs to be increased however.

Thank you, we have adopted these edits and the new figure will have larger fonts.

---

## Author Comment (AC2)

**Reviewer 2**

Climate model analyses have been limited to some extent by the scenarios used in projects such as CMIP6 and this study seeks to provide a framework for filling in some of the gaps left by the set of scenarios that exist. The authors perform a comprehensive evaluation of their framework primarily focussed on global mean temperatures and demonstrate its potential utility.

This study addresses an important issue and is a major contribution to the field. I only have minor comments for the authors to consider which I list below. I will admit that it took me a while to understand the methodology which isn't to fault the explanation given here, but I would suggest that the authors carefully read through the manuscript with a view to making the framework more easily understood where possible.

Thank you for the positive reaction, and we will definitely work on improving the readability, mindful of Reviewer 3's comments as well.

Minor comments:

L62-64: I agree that the SSP-RCPs span a range of forcings that probably covers the real-world outcome over this century but I think this sentence sounds a bit over-confident and could be dialled back a touch as "exhaustive" seems too strong a descriptor.

We have rephrased that as "well representative".

L71: Could also cite (Hawkins and Sutton 2009) as the paper where the method used in Lehner et al. originates.

We have added the citation.

L98: The focus on "transient" warming levels is introduced rather abruptly and I suspect the significance of this point may not be obvious to some readers. Perhaps a sentence or two explaining this could help. Papers that may be of use for an explanation include (Manabe et al. 1991; King et al. 2020; Callahan et al. 2021).

Thank you for the useful pointers. We have added a few words and the two more general studies as citations:

Here we extend this approach, which only produced isolated time windows, to the construction of entire transient scenarios, i.*e., a trajectory of greenhouse gases and other*

*anthropogenic forcings evolving continuously over the 21st century~\citep{Manabeetal1991,Kingetal2020}.*

L127: "dimension" should be "dimensions"

Thank you, corrected.

Figure 1: It might be worth reminding the reader either in the plot or caption that this is global mean temperature.

It is now specified (twice)  in the caption, thank you for the suggestion.

L227-228: Technically there is a lower bound of the level of global warming at the start of the simulations too presumably.

Yes, also in accordance to the discussion of Reviewer 1's comment we have more clearly described the range of applicability for STITCHES, considering the lower end challenge as well as the higher end, and overshoots and stabilization pathways.

L259: "do" should be "does"

Thank you, corrected.

L387-388: This sentence needs to be rewritten.

It now reads:. In this case STITCHES is significantly challenged, and its performance *as measured by the $E_r$ metric significantly diminished.*

L473-475: Remove "If" before "ENSO" and add "but" before "there exist".

Thank you, corrected.

L501: "haven't" should be "have not"

Thank you, corrected.

References

Callahan, C. W., C. Chen, M. Rugenstein, J. Bloch-Johnson, S. Yang, and E. J. Moyer, 2021: Robust decrease in El Niño/Southern Oscillation amplitude under long-term warming. Nat. Clim. Chang. 2021 119, **11**, 752–757, https://doi.org/10.1038/s41558-021-01099-2.

Hawkins, E., and R. Sutton, 2009: The potential to narrow uncertainty in regional climate predictions. Bull. Am. Meteorol. Soc., **90**, 1095–1107, https://doi.org/10.1175/2009BAMS2607.1.

King, A. D., T. P. Lane, B. J. Henley, and J. R. Brown, 2020: Global and regional impacts differ between transient and equilibrium warmer worlds. Nat. Clim. Chang., **10**, 42–47, https://doi.org/10.1038/s41558-019-0658-7.

Manabe, S., R. J. Stouffer, M. J. Spelman, and K. Bryan, 1991: Transient Responses of a Coupled Ocean–Atmosphere Model to Gradual Changes of Atmospheric CO2. Part I. Annual Mean Response. J. Clim., **4**, 785–818, https://doi.org/10.1175/1520-0442(1991)004<0785:TROACO>2.0.CO;2.

---

## Author Comment (AC3)

**Response to Reviewer 3**

This paper presents a procedure to create surrogate trajectories of climate model ensembles. The authors provide tests on a set of CMIP6 simulations and discuss the sensitivity to two key parameters of the procedure.

5   I have no reason to doubt that the authors know what they do. My main concern with the paper is that I neither understand the general picture nor the details.

>>We are sorry that our paper turns out to be so opaque to someone not familiar with the topic. We thank the >>reviewer for the open mindedness and fairness with which the paper was evaluated. We find the comments >>very useful and we have
10   made an effort to  to make our work better understandable by a larger audience.

My first concern is on the format of the paper and its suitability for ESD. The abstract, introduction, and conclusions are written by and for IPCC insiders, as the authors use a lot of IPCC jargon, which is obscure to most human beings, including me. This style of writing seems to go against the interdisciplinary nature of ESD. Not only the paper
15   does not report new understanding of the climate system, but the authors do not discuss that their procedure might help do so (or how). Another example is the use of the term "emulator" or "emulation". Of course, this remark is not limited to this manuscript. I yet have to see a reasonably clear definition of what is called a "climate emulator". For some authors, an emulator is a regression between some predictand
20   variable and a predictor. Here, it is obviously something else, that looks akin to analog modelling. Making a proper bibliographic search could help relate the procedure described in the manuscript to existing work, which might not appear in the IPCC reports. The notion of "creating new scenarios" is not clear. The IPCC seems to use SSP scenarios, which are relevant for the economy. What the authors do is
25   obviously something else. So, using this terminology might be confusing. The simple (acknowledged) fact that the emulation procedure cannot produce relevant GHG (or any forcing fluxes) should plead against the use of "creating scenarios". My understanding is that the procedure creates surrogate trajectories that are constrained by GSAT values. Why should those trajectories be called "scenarios" in
30   the IPCC sense?

>>These questions are extremely useful in pointing at the need for clarification and shedding jargon. We have >>attempted to do so, and in the process also clarified the emulator purpose, and its product. Throughout we >>have attempted to carefully phrase emulation as the emulation of ESM output, not emulation of scenarios, >>which is a contraption of the actual meaning that thanks to the reviewer comments we are now aware of >>being potentially confusing. You will find the revised manuscript extensively edited for clarity and when not >>completing doing away with it, defining the IPCC type nomenclature.

My second concern is that the procedure description seems inappropriately vague. Ideally, I should be able to reproduce the procedure by reading the manuscript (provided I have access to the data). The first step (l. 148) suggests that *one* time series of GSAT is created for each model by dumping together all ensembles, scenarios, etc. ([…] "the time series is made […]"). I guess/hope that the authors do differently. The fourth step (l. 157) is not clear: what is a target scenario? The authors allude to "target scenarios" in several places, but do not define what those are. I believe that the authors could design a diagram that explains how the procedure works. In practice, I understand that one needs to know the target scenario (i.e., have GSAT data). Hence, I do not understand how the authors can reconstruct "unknown" scenarios (e.g., SSP2) from just SSP1 and SSP5, which suggests that intermediate scenarios can be deduced from two extreme SSP scenarios. This might be true, but I would like to understand this miracle (at least for me).

>>We have evidently failed to communicate the basic set up of our problem. We have extensively rewritten the >>methods section, and we have now added a diagram that we hope should clarify the steps in the >>construction and the outcomes of the algorithm. Thank you for the suggestion of including such graphic.

My third major concern is on the results or the performance tests. The authors seem to be happy with the results reported in Figs 1-8. Indeed, the "emulated" time series are close to "targets" (whatever how the targets were designed). But is this desirable? The GSAT time series have no decadal or interdecadal variability (which might due to the procedure itself). This is not discussed, but I would doubt any procedure that

creates trajectories that do not yield long term variability are so useful, or really account for climate variability (e.g., the so-called butterfly effect). For me, the SOI results are "good" by construction, since they are excerpts of existing simulations. How would this emulation procedure be able to emulate changes in ENSO variability, which would be a key issue for impact modelling? My feeling is that the simulated trajectories give overconfidence about (the lack of) climate internal variability. The conclusion that this procedure can replace numerical model simulations hence seems overconfident.

>>Again, we think there is a fundamental problem of communication at play, as we are proposing a way to >>produce ESM output according to a new scenario stitching together existing ESM output, so STITCHES >>retains all the internal variability characteristics of the original ESM output. Of course, we are aware of and >>we discuss limitations in this regard, due to the idea of stitching together windows of existing simulations of >>9 year length, and due to the fundamental assumption that most variables are scenario independent in their >>behavior (this addresses the concern about ENSO variability changes) so their characteristics, as they would >>be produced by the ESM if running the scenario that instead is being emulated, are preserved as long as the >>algorithm matches the corresponding global warming levels. In particular, for ENSO variability changing at >>higher warming levels the idea is that we would sample such behavior for our emulation by sampling ESM >>output at high waring levels. Again, this assumes that the change in variability is essentially scenario >>independent and all that matters is the warming level.

**Minor issues**

In the search of nearest neighbors in the (T, dT) space (step 5, l. 165), are there different weights on T or dT, in the distance definition?

>>At the moment they are used with equal weight in a Euclidean distance, but the algorithm could be >>tuned/modified to define a distance that weighs one more than

the other. It's indeed an important >>possibility, and we have mentioned that in the text.

In step 6 (l. 170), what is a "pointer"?

>>We have rewritten extensively the description of the algorithm so we hope that now the terms will make >>better sense. We call pointer the identified archived experiment/time window that will give us content for a >>specific segment of our emulation.

l. 211: "(see 1)", what is "1"?

>>Apologies, the word "Table" was accidentally forgotten.

Figure 1: I can't read the labels on a printed version of the manuscript.

>>Also Reviewer 1 alerted us to the need of increasing the fonts. We will.

I feel that there should be a separate section that describes the experimental set ups, tests, etc.

>>We are hoping that by having clarified the rationale and functioning of the algorithm the present structure >>works better.

Figures 2-4: the captions should only keep descriptive statements, not comments that already appear in the text.

>>We have cleaned these up.

Equation (1) (l. 360): all symbols should be introduced. What is the bar? I think that $\hat y$ should be the synthetic and $y$ the truth, not the other way around, as suggested in l. 362. E2 is certainly not a ratio of variances (but a ratio of standard deviations). The denominator of E2 should be: $<(y - \bar y)^2>$ (no \hat).

>>Apologies, the reviewer is absolutely right that the equation and its terms are not rigorously presented and >>explained. We had also made a mistake in the formula,

where the denominator of the second component >>should be the same as that of the first, as the reviewer pointed out. We have fixed the typos, defined the >>quantities and symbols and corrected the text. We have also eliminated the bar notation, which had the >>same meaning as the angle brackets.

5    In conclusion, my feeling is that the manuscript would be much more appropriate in GMD, which incidentally has a better impact factor than ESD. Of course, this decision is left to the authors and the editor.

>>Thank you for helping us identify these shortcomings. We will attempt to make our paper clearer and >>hopefully more interesting to the audience of ESD, which
10   we would still prefer to pursue. In particular, we >>think the impact research community would be interested in our proposal, and we believe ESD could reach >>that community more easily than GMD.

---

## Author Response (AR2)

**Reviewer 1**

The authors have revised their manuscript and thoughtfully considered my comments and those of the other reviewers. In particular, the authors have made efforts to clarify their framework and added statements on its benefits and weaknesses. This is an important study which will hopefully initiate more discussion on climate modelling efforts in our community-congratulations!

*We thank Reviewer 1 for the thoughtful original comments and the positive reception towards our revision.*

**Reviewer 2**

I believe that the author's have answered most of my questions sufficiently and improved on the technical quality as well as readability of this paper. There are some areas that could still be elaborated/improved upon as follows:

L12: the term "construct ESM output" is misleading as you are constructing "ESM-like output" or otherwise "imitating/emulating" ESM output
*Fair point, we reworded as ESM-like output, as suggested.*

L101: when referring to Beusch et al. representing higher frequency fields, a more suitable reference is: https://doi.org/10.5194/esd-13-851-2022, same goes for L207 and L650
*We have substituted (first and last instance) and added (middle instance) this reference, thank you.*

Generally piece, segment as well as piecing and stitching seem to be used interchangeably and this could be streamlined.
*We have now tried to be consistent throughout. In particular, we have substituted 'pieces' for 'segments' in all cases (4) but for one instance where we say:*
*Our algorithm is applied separately to each individual ESM, as stitching together different models' **lengths of simulations** would almost certainly introduce spurious behavior.*
*We have only found one instance of the use of piecing, and we realized we were using it in an incorrect sense. We have now changed that into "splitting" as in:*
*the same smoothing and **splitting** procedure is applied to the trajectory of GSAT for the target scenario*

L210-L255: from reading other reviewer's comments I can imagine this is hard to follow for the general ESD audience. Figure A1 enriches this but could be made clearer. Some suggestions would be to put a legend for what the blue square vs yellow circle represent as well as numbering steps within the figure according to their corresponding step as outlined in the text.

In general, the steps in Figure A1 after the first middle blue box is hard to follow or match to the text and could be made clearer.

*Thank you for the feedback. We have followed both suggestions: we have now a legend at the bottom of the diagram explaining the difference between yellow circles and blue boxes and we have added numbering to the side of the boxes that refer to the respective steps as detailed in the itemized list of the "Methods" section.*

L340-L350: I appreciate the discussion of higher frequencies stitched together having a lower likelihood of introducing GSAT trajectory "jumps" due to high noise within the introduction section. The fact that this is being demonstrated here should be made more explicit however, additionally what are the GSAT's recovered from? monthly or yearly gridded temperature?

*We have added a parenthetical clause to clarify that our GSAT time series are made of annual values. The paragraph now starts as:*
*For all cases when the emulation of GSAT time series (made of annual average values) […]*

*We have also added a short paragraph at the end of the section following the reviewer suggestion to stress the results of this validation of higher frequency, noisier quantities:*

*On the basis of these results, we confirm the correctness of our expectation that, after validating the statistical characteristics of a large scale, low frequency quantity like annual GSAT, further validation of emulated variables at grid-point scale and higher temporal frequency do not seem to present larger challenges. The higher noise of these quantities indeed accommodates the discontinuities introduced by their emulation.*

L415: a good reference for ice-free summers could be: https://doi.org/10.1175/JCLI-D-15-0284.1

*We have added this reference, thank you.*

Figure 5: The term monthly trends remains a bit ambiguous here: is it a month-specific trend, in which case the month should be specified in the title/caption. If not how was this calculated? Since e.g. winter monthly warm quicker than summer, one would expect that this would be taken into account here and if not why?

*We have added the following text when describing these results:*
*All metrics here are computed using time series of gridded output at monthly frequency, covering the entire annual cycle, for the length of the emulated output (2015-2100). In the appendix we show similar results for month-specific output sampling behavior during Boreal winter (January) and Boreal summer (July), addressing the possibility that the emulation could be differently challenged by stronger or weaker forced trends.*

Figure D7: nice addition! *Thank you*

*We thank reviewer 2 for the thoughtful original comments and the positive reception of our revised version, together with these additional points for improvement that we hope to have addressed as detailed above.*

**Reviewer 3**

I thank the authors for the clarifications they brought after my admittedly candid comments on the original manuscript. I now have a better feel of the STITCHES procedure.

*We thank the reviewer for giving the study another try. We feel as if there is still a fundamental challenge in the communication of our algorithm's purpose and uses. We have tried to respond to the reviewer's comments here, but we could not really find a justification to modify our text and figures, as a consequence. We also think some of our text and figures have been misinterpreted but given that the other reviewers do not raise issues, in fact in some instances are openly happy with our figures we would ask not to be made to change what we have included.*

Major comments
I am a bit skeptical about the interest for climate scientists of the paper. Unless I missed it in the discussion section, there is no mention of internal climate variability that makes ensemble members (of the same model) yield very different behavior (e.g. Deser et al. Nat. Clim. Chang. 10, 277–286 (2020). https://doi.org/10.1038/s41558-020-0731-2). I think that the physical caveats should be discussed in the light of the literature on climate variability, otherwise I feel that results reported in Figure 2—4 are likely to bring overconfidence in climate projections to impact communities.

*We think that the interest by climate scientist in our solution will come from its potential in lightening the burden of running many similar scenarios, a demand that comes from the impact modeling research community. Please, see later for more on this point.*

*As for STITCHES and its emulation of internal variability (IV), that is indeed a very important aspect of emulation that we seek to address, when we set out to emulate high frequency ESM output. Were we not concerned with IV, simple pattern scaling of multidecadal mean behavior could still be used as a solution to emulation. However, as we describe in the paper, state of the art impact models require realistic sequences of relatively high frequency quantities, and modelers are aware of the need of addressing the interplay of forced changes and IV. Hence the need for an emulator that preserved aspects of IV, besides the forced component of climate change.*

*In the paper, after validating aspects of IV specific to a single realization we specifically validate the emulation of initial condition ensembles. This is one of two specific uses of the emulator, and we have dedicated a subsection to its validation. We introduced the metric Er, one component of which, E2, addresses the accuracy with which the behavior of the emulated ensemble mimics that of the real one. Table 4 and Section 3.1.2 describe these results.*

*Also, by construction, the algorithm is careful not to repeat pieces across the initial condition ensemble members emulated, in order not to create identical behavior across ensemble members.*

*Last, our algorithm replicates, again by construction, the internal variability of an ESM within the pieces used in the stitching. Of course, also by construction, and as we discuss, characteristics of internal variability that go beyond the 9-year window used for our pieces (i.e., constituting the building blocks) won't be emulated faithfully, and we clearly describe this among the caveats, when adopting STITCHES output rather than true ESM output.*

One un-discussed issue of the paper is climate sensitivity: since STITCHES is based on GSAT variations, one could in principle test time series of CO2 atmospheric concentrations, which are available in ESM simulations with ssp scenarios. Is it possible that STITCHES violates the underlying climate sensitivities for each model? This should be fairly easy to assess, and might interest the communities who deal with biogeochemical cycles.

*We apologize, but we do not understand what the reviewer is proposing here. We show in our validation section that trends, specifically trends in temperature, and century-long trends in particular, are preserved for the individual models emulated. That would suggest that STITCHES does not modify/violate the climate sensitivity of the emulated ESM in any significant way (again, for the purpose of impact modeling). We are not sure how impact modeling would be concerned with climate sensitivity per-se, beyond the need to choose ESMs that span its assessed range, which the CMIP6 models used here, and against which we validate our algorithm successfully, surely do.*

Specific comments

I am still puzzled by some statements and figures of the revised manuscript.

Figures 2-4 do not look very informative, at least from the statistical point of view, as the lines all overlap each other, and all look undistinguishable. Is there a case where stitched and target time series do not look alike? Could all the panels be summarized in just one figure? (See my first major point)

*We do not intend these figures to substitute for a rigorous validation, which we describe extensively in the text. We wanted to present the reader with the range of models/ensembles that we set out to emulate, and give a visual impression of the results, leaving to the text the details of our extensive validatory procedures. As for cases where stitched and target time series do not look alike, it is generally a result of selecting a $Z$ value (tolerance for the search of nearest neighbors) that is too large, and represents an emulation failure. Rather than plot the results of failure, which may take many forms, we devote a section of our results to documenting our estimation of $Z_{cutoff}$, the value at which the generated ensemble is statistically consistent with the target ensemble and therefore appear 'indistinguishable' (i.e. clearly following the same trend as the target and displaying consistent internal variability).*

The ENSO section is strange and very qualitative: from the described procedure, would there be a possibility that the emulated versions DO NOT look like the original versions? In other words, is it not by construction, from the reshuffling procedure, that ENSO time series yield

realistic features, at least visually? The auto-correlation figures (Fig. 8) are not particularly convincing. In particular, those graphs do not show frequencies, as claimed in the text (l. 397). Why is the auto-correlation (ACF) equal to 0 at lag 0 for the CAMS model? The ACF at lag 0 is the variance, and should NEVER be zero!

*We wanted to once again give a visual impression with the first type of plots, and then substantiate our claims with the temporal and spectral characteristics of the time series (we added a figure for that purpose following Reviewer 2's suggestion, which they appreciated). The 0 values at lag 0 are seen in the plots of the PARTIAL autocorrelation functions (**P**ACFs, second and fourth rows of plots) not of the autocorrelation functions (ACFs, first and third rows of plots). The PACF is defined as the partial autocorrelation of the values of a time series with its own lagged values, and therefore is defined for lags 1 and higher. The plots of the ACF show the taller spike at lag 0, as is to be expected, indeed representing the variance of the time series. Also, we are not claiming that the ACF/PACF graphs show frequencies. Assuming the reviewer does not agree with the use of the term "densities" we are referring to the additional figure D7 in the appendix, which shows spectral densities (requested and now approved by Reviewer 2).*

Could the authors give an illustrated example of the utility of STITCHES to climate scientists? (see my second major point).
*STITCHES is a tool conceived to support impact modeling, first and foremost. Its utility to climate scientists resides in "freeing up" the resources that are right now dedicated to running many slightly different scenarios, or many members of initial condition ensembles. Particularly, our figure 1 highlights a view of ESM outputs that may be valuable for CMIP7 scenario planning as climate scientists consider trade-offs among allocation of computational resources to different type of experiments (scenarios and other types of simulations). We do not claim that our emulator has value in uncovering physical behaviors that a climate scientist would find novel. What we confirm from our validation exercises has been well known for the last few years at least: many impact-relevant variables are "slaved to GSAT", and not path dependent. This is an interesting result from the climate science research community that we exploit in patching together our new scenarios/ensemble members on the basis of the corresponding behavior of GSAT.  Thus, we could claim that STITCHES allows the climate science research community to benefit from their own finding in a concrete, resource-preserving way in future exercises.*

---

## Author Response (AR3)

Dear Prof. Messori,

Thank you for your positive reception of our revised article and the further input.
We have used all your suggestions in the final version that we have now uploaded.

In particular, as can be gauged by the "diff" file we include with this response:

1. *With reference to one of the comments of Reviewer #2, you may want to doube-check again all instances of "ESM output" vs. "ESM-like output". For example, on ll. 16 and 33 it may be more appropriate to speak of "ESM-like output", as you are referring to the emulator data. There may be other such instances later in the text.*
We have systematically searched for every instance of the use of the word "output" and corrected the expression "ESM output" to ESM-like output in all instances when we refer to the emulator results.

2. *With reference to one of the comments from Reviewer #3. Although this is to some extent subjective, you could evaluate whether showing 18 panels of coloured/black "spaghetti" in Figs. 2-4 is truly informative for readers, or whether you could condense these figures in a single one (e.g. pick models with high/low sensitivity or high/low internal varability for the two scenarios as illustrative examples) and move the rest of the panels to the Appendix.*
We have kept the first figure in the main text, which happened to show three ESMs with three different archive size and three different ECS values, and we have moved the other two figures in the appendix with the rest. We have modified text and caption accordingly.

3. *Regarding the ENSO section, it may be worth mentioning that PACF is not defined at lag 0, as in some programming languages (and hence plots in other studies) PACF at lag 0 is defined as ACF at lag 0. In the same section, I believe the Reviewer was questioning the use of the term "frequencies" in the passage: "note however that we are not comparing these frequency characteristics to observations, which is not the point of our validation exercise). We consider this validation particularly important, both because of the salience of ENSO behavior for many types of impact, and because the frequency characteristics of this mode of variability are close to our 9-year windows." You may want to specify that the validation is performed using (P)ACF and spectral densities.*
We have added a noter to each caption (in the main text and appendix) under the figures showing ACFs/PACFs specifying that the R function we use to compute the latter does not define it for lag 0. Also, in the text we have added a sentence that explains that we validate the time series behavior through (P)ACFs and spectral densities as you suggest.

*4. I quite liked the explanation you provided to the final comment of Reviewer #3, stating that "many impact-relevant variables are "slaved to GSAT"". This provides a clear, easy to understand statement. You may want to consider including something similar in your main text, as a perhaps somewhat colloquial but clear and simple statement for the non technically-minded reader.*

Thank you for this suggestion, we have added words to a sentence in the second paragraph of the method session to this effect that now reads: "If the stitching works for GSAT, we show that we can also stitch together the corresponding pieces of simulations for many other impact-relevant variables that are in essence slaved to GSAT, at a range of time and spatial scales, without introducing artifacts and discontinuities of consequence for most application in impact research, especially in the context of the uncertainties that climate or impact models are well known to introduce.

We hope this makes the new version satisfactory.

Thank you again for your fair handling of our manuscript.

Claudia, Abigail and Kalyn

[revised manuscript text omitted]